# Spectral properties, localization transition and multifractal eigenvectors of the Laplacian on heterogeneous networks

**Jeferson D. da Silva[1]\*, Diego Tapias[2,†], Peter Sollich[2,3‡] and Fernando L. Metz[1°]**

**1** Physics Institute, Federal University of Rio Grande do Sul, 91501-970 Porto Alegre, Brazil
**2** Institute for Theoretical Physics, Georg-August-Universität Göttingen, Göttingen, Germany
**3** Department of Mathematics, King's College London, London WC2R 2LS, UK

\* jeferson.silva@ufrgs.br, † diego.tapias@theorie.physik.uni-goettingen.de, ‡
peter.sollich@uni-goettingen.de, ∘ fmetzfmetz@gmail.com

## Abstract

We study the spectral properties and eigenvector statistics of the Laplacian on highly-connected networks with random coupling strengths and a gamma distribution of rescaled degrees. The spectral density, the distribution of the local density of states, the singularity spectrum and the multifractal exponents of this model exhibit a rich behaviour as a function of the first two moments of the coupling strengths and the variance of the rescaled degrees. In the case of random coupling strengths, the spectral density diverges within the bulk of the spectrum when degree fluctuations are strong enough. The emergence of this singular behaviour marks a transition from non-ergodic delocalized states to localized eigenvectors that exhibit pronounced multifractal scaling. For constant coupling strengths, the bulk of the spectrum is characterized by a regular spectral density. In this case, the corresponding eigenvectors display localization properties reminiscent of the critical point of the Anderson localization transition on random graphs.

# 1  Introduction

One crucial aspect of complex systems is the heterogeneous structure of the interactions among their constituents [1]. This heterogeneity manifests itself in two primary forms. First, the local neighbourhood around each unit may exhibit a random topology of connections, resulting in a variable number of neighbours per node. Second, the magnitudes of interactions among the units may themselves be random variables. The interplay between heterogeneous topology and random coupling strengths gives rise to a variety of interesting dynamical phenomena [2].

The spectral properties of matrices associated with networks are key to understanding how the heterogeneous structures of complex systems shape dynamical processes occurring on them. The adjacency and the Laplacian matrix are the most prominent examples of matrices constructed from networks [3]. The adjacency matrix encodes the pairwise interactions among the units of the underlying network. Its spectral properties determine, among other things, the stability of large complex systems subject to small external perturbations [4–6]. The Laplacian is constructed as the difference between the adjacency matrix and a diagonal matrix [3], whose elements are chosen such that all row and column sums are zero [7,8]. This row/column constraint naturally arises in various contexts due to conservation laws, such as conservation of linear momentum or probability [9]. The statistical properties of the eigenvalues and eigenvectors of the Laplacian determine the solution of several problems on networks, including the long-time behaviour of diffusion and of search algorithms [10], the nature of vibrational modes in disordered systems [9,11], the spectrum of relaxation rates in trap models of glassy dynamics [12–15], the distribution of resonances in random impedance networks [16,17], the dynamics of consensus algorithms [18,19], and the stability of the synchronized phase in oscillator networks [20,21].

Since the seminal work of Rodgers and Bray [22], the problem of determining the spectral properties of the Laplacian on heterogeneous networks has attracted enormous interest [11, 16, 17, 22–29]. In the case of fully-connected networks, where each node is coupled to all others, the pairwise coupling strengths are typically random, but the absence of fluctuations in the local topology renders the network structure homogeneous, simplifying the problem. The spectral density or equivalently the eigenvalue distribution of the Laplacian follows in this case from the solution of a single equation for the average resolvent, which can be derived using both the replica method [22] and the supersymmetry approach [16,17]. A rigorous proof for the spectral density of the Laplacian on fully-connected networks has been established in [28].

The picture is radically different in the case of sparse networks, where each node is connected on average to a finite number $c$ of neighbours. Alongside the stochastic nature of the coupling strengths, models defined on sparse networks may have a heterogeneous topology

because the number of neighbours of each node, commonly referred to as its degree, is generally a random variable. The prototypical network model for studying the role of fluctuating degrees in a controllable manner is the configuration model [30, 31], in which the degree distribution is specified from the outset. The general formalism for the spectral density of the Laplacian on sparse networks has been developed in a series of papers [11, 22, 24–27], culminating in a pair of self-consistent equations for the full distribution of the diagonal elements of the resolvent matrix [25–27], which have been rigorously demonstrated in [32]. The numerical solutions of these so-called resolvent distributional equations form the basis for understanding how network heterogeneity impacts the spectral and localisation properties of the Laplacian, providing insights not only into the spectral density [25, 26] but also into the statistical properties of the eigenvectors [27]. However, despite their pivotal role, the resolvent equations admit analytic solutions only in the homogeneous regime, where the relative variance of the degree distribution is zero and the resolvent elements are independent of the node.

In a series of recent works [33–35], it has been demonstrated that the resolvent equations for the *adjacency matrix* of heterogeneous networks do allow for the derivation of analytic solutions in the high-connectivity limit $c \to \infty$, as long as the relative variance of the degree distribution remains finite. In this intermediate connectivity regime, lying between sparse and fully-connected networks, the resolvent distributional equations simplify into a single self-consistent equation for the average resolvent, which explicitly incorporates the effect of degree fluctuations. When considering a negative binomial distribution of degrees, the average resolvent solely depends on two control parameters: the variance of coupling strengths and the relative variance of the degree distribution [33–35]. This setting enables a thorough investigation of how network heterogeneity impacts the spectral properties and the eigenvector statistics of the adjacency matrix, since one can independently control fluctuations in both network topology and coupling strengths.

While the findings in [33–35] apply to the high-connectivity limit, they unveil a surprisingly rich phenomenology driven by network heterogeneities. Specifically, strong degree fluctuations lead to a divergence within the bulk of the spectral density. This singular behaviour is caused by the power-law decay observed in the distribution of the imaginary part of the resolvent [34], a consequence of the strong spatial fluctuations of the eigenvectors and their localised behaviour [35]. The results for adjacency matrices in [33–35] hint at the potential existence of nontrivial solutions to the resolvent equations of other random matrices in the high-connectivity regime. This could pave the way to a systematic exploration of the role of network heterogeneities in the spectral properties of other important matrices. The *Laplacian* is a prime candidate for probing such solutions, given its significance in governing dynamical processes on networks.

In this work, we obtain the solution of the resolvent equations for the Laplacian on heterogeneous networks in the high-connectivity regime. Unlike the Laplacian on fully-connected networks, the statistics of the resolvent elements are non-universal, as they explicitly depend on the distribution of the degrees rescaled by their mean $c$. Building on previous works [33–35] and considering a gamma distribution of rescaled degrees, the analytical results are expressed solely in terms of the first two moments of the coupling strengths as well as the variance $1/\gamma$ of rescaled degrees. This framework enables a continuous interpolation between the two extremes of pure heterogeneous topology and pure heterogeneous coupling strengths ($\gamma \to \infty$). We present a number of analytical findings and a thorough analysis of the spectral density, the distribution of the local density of states (LDoS), the singularity spectrum and the multifractal exponents, forming a comprehensive picture of the spectral properties and the eigenvector statistics of the Laplacian on heterogeneous networks.

We separate our results into two different cases depending on the nature of the coupling

strengths. For homogeneous couplings, the bulk of the eigenvalue distribution is described by a regular function, which becomes singular only at the lower spectral edge. The LDoS distribution for states within the bulk is characterised by a singular behaviour and a power-law decay with exponent **3/2**. These are typical properties of the localized phase [36–38]. In fact, by computing the singularity spectrum and the multifractal exponents, we confirm that all eigenvectors are localized for any finite value of $\gamma$. Interestingly, these spectral observables are identical to those found at the critical point of the Anderson localization transition on random graphs [38–40]. In the more complex scenario of heterogeneous coupling strengths, the spectral density diverges in the bulk of the spectrum (at eigenvalue zero) as long as $\gamma \leq 1/2$. The LDoS distribution at this zero eigenvalue is a regular function that decays as a power-law with exponent $\gamma + 3/2$. The appearance of the singularity in the spectral density is accompanied by a delocalization-localization transition as a function of the variance $1/\gamma$ of rescaled degrees. By computing the singularity spectrum and the multifractal exponents of the corresponding eigenvectors, we provide compelling evidence that, for $\gamma > 1/2$, the eigenvectors are extended and non-ergodic, while for $\gamma < 1/2$ they become localised, with multifractal exponents that depend on the strength of degree fluctuations. Remarkably, the localization in both cases is an instance of the *statistical localization* mechanism introduced in Ref. [35], whereby the localization is correlated to a single-node statistical property (e.g. the degrees), instead of being related to the spatial structure of the underlying network.

The rest of the paper is organized as follows. In the next section, we introduce the Laplacian matrix on networks and the spectral observables that are the focus of our interest in this work. Section 3 presents our main analytical expressions for the spectral observables in the high-connectivity regime. These expressions are derived by taking the limit $c \to \infty$ in the resolvent distributional equations of sparse networks with arbitrary distributions of degrees and coupling strengths. The analytical results of section 3 enable a systematic study of the combined role of topological disorder and random coupling strengths in the spectral observables. In section 4, we present analytical and numerical results for a gamma distribution of rescaled degrees. The final section summarizes our work and discusses a number of interesting avenues for future research. Supplementary material can be found in three appendices. In appendix A, we explain in detail how to take the limit $c \to \infty$ of the resolvent equations; in appendix B, we derive the distribution of squared eigenvector components for homogeneous coupling strengths; while appendix C provides details of the numerical calculation of the singularity spectrum and the multifractal exponents.

## 2    The Laplacian matrix of networks and its spectral observables

We consider a simple and undirected random graph with $N$ nodes [41]. The graph structure is determined by the set of binary random variables $\{c_{ij}\}$ $(i, j = 1, \ldots, N)$, where $c_{ij} = c_{ji} = 1$ if there is an undirected edge $(i, j)$ connecting nodes $i$ and $j$ $(i \neq j)$, and $c_{ij} = 0$ otherwise. We set $c_{ii} = 0$ and further associate a symmetric coupling strength $J_{ij} = J_{ji} \in \mathbb{R}$ to each edge $(i, j)$. The degree $k_i = \sum_{j=1}^{N} c_{ij}$ is a random variable that specifies the number of nodes adjacent to $i$. The degree distribution $p_k$ and the average degree $c$ are defined, respectively, as

$$p_k = \lim_{N \to \infty} \frac{1}{N} \sum_{i=1}^{N} \delta_{k,k_i}, \quad c = \sum_{k=0}^{\infty} k p_k. \tag{1}$$

The degree distribution, which is one of the primary quantities characterizing the network structure, gives the probability that a randomly chosen node has degree $k$.

An instance of the random network is completely specified by the random variables $\{c_{ij}\}$ and $\{J_{ij}\}$. We assign the entries $\{c_{ij}\}$ of the adjacency matrix from the configuration model

[30, 31], in which a single graph instance is uniformly sampled from the set of all possible graphs with a prescribed degree sequence $k_1, \ldots, k_N$, generated from $p_k$. The configuration model provides the ideal platform for investigating how local network heterogeneities impact the spectral properties, since the entire degree distribution acts a control parameter of the model.

We take the coupling strengths $\{J_{ij}\}$ as i.i.d. real variables drawn from some distribution $p_J$ whose mean and variance are given, respectively, by $J_0/c$ and $J_1^2/c$[1]. In addition, we assume that higher-order moments of $p_J$ are proportional to $1/c^\beta$ with $\beta > 1$. This scaling ensures that the spectral observables converge to a finite limit as $c \to \infty$.

We are interested in the spectral properties of the Laplacian matrix $L$ associated with the undirected network defined by $\{c_{ij}\}$ and $\{J_{ij}\}$. The entries of the $N \times N$ Laplacian read

$$L_{ij} = \delta_{ij} \sum_{k=1}^{N} c_{ik} J_{ik} - c_{ij} J_{ij}. \tag{2}$$

The non-diagonal terms in Eq. (2) are the elements of the adjacency matrix weighted by the coupling strengths $J_{ij}$, while the form of $L_{ii}$ ensures that the constraints $\sum_{j=1}^{N} L_{ij} = \sum_{j=1}^{N} L_{ji} = 0$ are fulfilled. The matrix $L$ has a complete set $\{\vec{\psi}_\mu\}_{\mu=1,\ldots,N}$ of orthonormal eigenvectors that satisfy

$$L\vec{\psi}_\mu = \lambda_\mu \vec{\psi}_\mu, \tag{3}$$

where $\{\lambda_\mu\}_{\mu=1,\ldots,N}$ represents the set of (real) eigenvalues of $L$. The condition $\sum_{j=1} L_{ij} = 0$ implies that $L$ has a single eigenvalue $\lambda_1 = 0$ with a corresponding uniform eigenvector $\vec{\psi}_1^T = \frac{1}{\sqrt{N}}(1, \ldots, 1)$. We assume here that the network consists of a single connected component, which is always the case in the limit of large degrees $c \to \infty$ considered later. Note that while the standard Laplacian – constructed without the weights $J_{ij}$ – is positive semi-definite [3], here $L$ can have negative eigenvalues because the couplings $J_{ij}$ are real-valued. The spectral properties of the weighted Laplacian, with a mixture of positive and negative couplings, find applications in the study of consensus dynamics [18, 19], impedance networks [16, 17], and instantaneous normal modes in liquids [11].

The simplest quantity that characterizes the spectrum of $L$ is the empirical *spectral density*

$$\rho(\lambda) = \lim_{N \to \infty} \frac{1}{N} \sum_{\mu=1}^{N} \delta(\lambda - \lambda_\mu), \tag{4}$$

also known as the *density of states* (DoS) in the context of condensed matter physics. An analogue of $\rho(\lambda)$ that is resolved across the nodes of the network and gives information on the statistics of the eigenvector components is the *local density of states* (LDoS) [42]

$$\rho_i(\lambda) = \sum_{\mu=1}^{N} |\psi_{\mu,i}|^2 \delta(\lambda - \lambda_\mu), \tag{5}$$

where $\psi_{\mu,i}$ is the component of $\vec{\psi}_\mu$ at node $i$. The LDoS gives the overall contribution at node $i$ from the eigenvector intensities $|\psi_{\mu,i}|^2$ for eigenvalues around $\lambda$. Since in general $\rho_i(\lambda)$ fluctuates from site to site, it is useful to introduce the empirical distribution of $\rho_i(\lambda)$,

$$P_\lambda(y) = \lim_{N \to \infty} \frac{1}{N} \sum_{i=1}^{N} \delta(y - \rho_i(\lambda)), \tag{6}$$

---

[1]The variance of $p_J$ has a subleading contribution of $\mathcal{O}(1/c^2)$ in the limit $c \to \infty$.

which characterizes the spatial fluctuations of the eigenvector components corresponding to eigenvalues around $\lambda$. Clearly, the empirical spectral density in Eq. (4) is the first moment of $P_\lambda(y)$.

A more comprehensive way of characterizing the spatial distribution of the eigenvector components consists in determining the set of *multifractal exponents* and the corresponding *singularity spectrum*. For a given eigenvalue $\lambda$, let $|\psi_i|^2$ be the squared amplitude of the corresponding eigenvector at node $i$. The quantities $|\psi_1|^2, \dots, |\psi_N|^2$ define a normalized distribution across the nodes of the network. This distribution can be characterized by its moments (also called generalized inverse participation ratios [38, 43])

$$I_q(N) = \sum_{i=1}^{N} |\psi_i|^{2q}. \tag{7}$$

For large $N$, the statistical properties of the $|\psi_i|^2$ are encoded in the set of multifractal exponents $\tau(q)$, which one introduces via the scaling relation [38, 43–45]

$$I_q(N) \sim N^{-\tau(q)}. \tag{8}$$

The function $\tau(q)$ is convex up, nondecreasing, and it satisfies the conditions $\tau(0) = -1$ and $\tau(1) = 0$. An alternative way of capturing the spatial fluctuations of $|\psi_i|^2$ is by defining a scaling exponent at each node $i$ through the scaling form $|\psi_i|^2 \sim N^{-\alpha_i}$ ($\alpha_i \geq 0$). For large $N$, the empirical distribution of $\alpha_1, \dots, \alpha_N$, formally defined as

$$\Omega(\alpha) = \frac{1}{N} \sum_{i=1}^{N} \delta(\alpha - \alpha_i), \tag{9}$$

then has the scaling form [43, 45]

$$\Omega(\alpha) \sim N^{f(\alpha)-1}, \tag{10}$$

where the function $f(\alpha)$ is the so-called singularity spectrum (or spectrum of fractal dimensions). The singularity spectrum is a convex up function of $\alpha \geq 0$ with a maximum value of unity. Equations (7-10) imply that the multifractal exponents and the singularity spectrum are related via a Legendre transform. To show this, one rewrites the $q$-th moment $I_q(N)$ for large $N$ as

$$I_q(N) = \sum_{i=1}^{N} N^{-\alpha_i q} = N \int \Omega(\alpha) N^{-\alpha q} d\alpha \overset{N \gg 1}{\sim} \int N^{f(\alpha)-\alpha q} d\alpha. \tag{11}$$

By solving the integral using a saddle-point approximation and then comparing the result with Eq. (8), one finds the Legendre transform

$$-\tau(q) = \max_\alpha [f(\alpha) - q\alpha]. \tag{12}$$

The singularity spectrum is also directly connected to the distribution of $|\psi_i|^2$ [39, 46]. In order to show this relation, let us introduce the rescaled variable $x_i = N|\psi_i|^2$ and its empirical distribution $P_\psi(x)$. Since $|\psi_i|^2 \sim N^{-\alpha_i}$ for large $N$, we can perform the change of variables $x \sim N^{1-\alpha}$ and rewrite the distribution $P_\psi(x)$ in terms of $\Omega(\alpha)$. By combining the resulting expression with the scaling assumption of Eq. (10), it follows that

$$P_\psi(x) = \frac{A}{x \ln N} N^{f(\alpha)-1}, \tag{13}$$

where $A$ is an $N$-independent constant of order unity. The above expression holds for large $N$. In the limit $N \to \infty$, Eq. (13) can be simplified to

$$f(\alpha) = \lim_{N \to \infty} \frac{\ln(x N P_\psi(x))}{\ln N}. \tag{14}$$

Thus, the singularity spectrum can be computed from the spatial distribution of the (scaled) squared amplitudes $x_i = N|\psi_i|^2$ in the limit $N \to \infty$.

The statistical properties of both the eigenvalues and the eigenvectors of $L$ are encoded in the $N \times N$ resolvent matrix [23, 27, 47, 48]

$$G(z) = (Iz - L)^{-1}, \tag{15}$$

where $z = \lambda - i\epsilon$ ($\epsilon > 0$) lies in the complex lower half-plane and $I$ denotes the identity matrix. The diagonal elements $\{G_{ii}(z)\}_{i=1,...,N}$ of $G(z)$ determine the regularized forms of the local density of states [42]

$$\rho_i(z) = \frac{1}{\pi} \mathrm{Im}\, G_{ii}(z) \tag{16}$$

and of the spectral density

$$\rho_\epsilon(\lambda) = \frac{1}{\pi} \lim_{N \to \infty} \frac{1}{N} \sum_{i=1}^{N} \mathrm{Im}\, G_{ii}(z). \tag{17}$$

The original quantities, $\rho_i(\lambda)$ and $\rho(\lambda)$, are obtained by taking the limit $\epsilon \to 0^+$ in Eqs. (16) and (17), respectively. In addition, correlation functions between the resolvent elements determine the generalized inverse participation ratios [23, 47]. Thus, it is convenient to introduce the joint empirical distribution $\mathcal{P}_z(g) \equiv \mathcal{P}_z(\mathrm{Re}\, g, \mathrm{Im}\, g)$ of the real and imaginary parts of $G_{ii}(z)$,

$$\mathcal{P}_z(g) = \lim_{N \to \infty} \frac{1}{N} \sum_{i=1}^{N} \delta(g - G_{ii}(z)) \tag{18}$$

since this object gives access to the spectral and localization properties of $L$. The support of $\mathcal{P}_z(g)$ lies in the complex upper half-plane $\mathbb{H}^+$. We are interested in the empirical distribution $\mathcal{P}_z(y)$ of the imaginary part $y_i = \mathrm{Im}\, G_{ii}(z)$ of the resolvent, obtained by marginalizing $\mathcal{P}_z(g)$ with respect to $\mathrm{Re}\, g$. The first moment of $\mathcal{P}_z(y)$ yields the spectral density, while the full distribution $\mathcal{P}_z(y)$ is essentially the distribution of the LDoS, which provides valuable information about the statistics of the $|\psi_i|^2$.

## 3 The high-connectivity limit of the spectral observables

We begin this section by explaining the central idea of the cavity method and how this approach leads to a pair of distributional equations for $\mathcal{P}_z(g)$ in the case of sparse networks, where the average degree $c$ remains finite as $N \to \infty$. In the second part of this section, we discuss how to take the limit $c \to \infty$ in the distributional equations and thus calculate $\mathcal{P}_z(g)$ in the high-connectivity limit. The details of the calculation are presented in appendix A.

The cavity approach to random matrices relies on the local tree-like structure of the underlying sparse network constructed from the adjacency matrix [48, 49]. Let us consider a single graph instance drawn from the configuration model with a large number $N$ of nodes and finite $c$. The local tree-like property means that the probability of finding short loops at a finite distance from a randomly chosen node vanishes as $N \to \infty$ [32]. As a consequence, the resolvent elements in the neighbourhood of an arbitrary node $i$ are correlated essentially only

through $i$, thanks to the tree-like structure around the node in question. This key property allows us to write the diagonal elements of the resolvent $G(z)$, for a single graph instance, as follows [27]

$$G_{ii}(z) = \frac{1}{z - \sum_{j \in \partial_i} J_{ij} \left[ 1 - J_{ij} G_{jj}^{(i)}(z) \right]^{-1}}, \tag{19}$$

with $\partial_i$ denoting the set of nodes adjacent to $i$. The complex variable $G_{jj}^{(i)}(z)$ is the $j$th-diagonal element of the resolvent matrix on the so-called cavity graph, which is a graph where node $i \in \partial_j$ and all its edges have been removed. The cavity variables $\{G_{jj}^{(i)}\}$ are determined by the fixed-point equations

$$G_{jj}^{(i)}(z) = \frac{1}{z - \sum_{\ell \in \partial_j \setminus i} J_{j\ell} \left[ 1 - J_{j\ell} G_{\ell\ell}^{(j)}(z) \right]^{-1}} \quad (i \in \partial_j), \tag{20}$$

with $\partial_j \setminus i$ denoting the set of nodes adjacent to $j$ excluding $i$. Equations (19) and (20) become asymptotically exact as $N$ grows to infinity [32]. In this regime, it is more convenient to work with the empirical distributions of $G_{ii}(z)$ and $G_{jj}^{(i)}(z)$. Since both sides of Eq. (19) are equal in distribution, $\mathcal{P}_z(g)$ is determined by

$$\mathcal{P}_z(g) = \sum_{k=0}^{\infty} p_k \int_{\mathbb{H}^+} \left[ \prod_{\ell=1}^{k} dg_\ell \, \mathcal{Q}_z(g_\ell) \right] \int_{\mathbb{R}} \left[ \prod_{\ell=1}^{k} dJ_\ell \, p_J(J_\ell) \right] \delta \left[ g - \frac{1}{z - \sum_{\ell=1}^{k} J_\ell (1 - J_\ell g_\ell)^{-1}} \right], \tag{21}$$

where $\mathcal{Q}_z(g)$ is the joint empirical distribution of the real and imaginary parts of $G_{jj}^{(i)}(z)$. The function $\mathcal{Q}_z(g)$ solves the self-consistent equation

$$\mathcal{Q}_z(g) = \sum_{k=1}^{\infty} \frac{k}{c} p_k \int_{\mathbb{H}^+} \left[ \prod_{\ell=1}^{k-1} dg_\ell \, \mathcal{Q}_z(g_\ell) \right] \int_{\mathbb{R}} \left[ \prod_{\ell=1}^{k-1} dJ_\ell \, p_J(J_\ell) \right] \delta \left[ g - \frac{1}{z - \sum_{\ell=1}^{k-1} J_\ell (1 - J_\ell g_\ell)^{-1}} \right]. \tag{22}$$

Equations (21) and (22) can also be derived using the replica method [25, 26] of disordered systems. Although these equations play a pivotal role in determining the spectral observables of sparse random graphs with arbitrary $p_k$, they in general admit exact closed form solutions only in the absence of disorder, i.e. when all degrees and coupling strengths are identical $(p_k = \delta_{k,c}, \, p_J(J_\ell) = \delta(J_\ell - J))$.

In appendix A, we explain how progress beyond this rather limited scenario can be made: we calculate there the spectral observables of interest in the high-connectivity limit and for an arbitrary degree distribution $p_k$ by taking the limit $c \to \infty$ in Eqs. (21) and (22). The empirical spectral density $\rho_\epsilon(\lambda)$ and the distribution $\mathcal{P}_z(g)$ of the resolvent entries are given, respectively, by

$$\rho_\epsilon(\lambda) = \frac{1}{\sqrt{2\pi J_1^2}} \operatorname{Re} \left[ \int_0^\infty d\kappa \, \kappa^{-1/2} v(\kappa) \, w \left( \frac{\kappa J_0 + \kappa J_1^2 \langle G \rangle - z}{\sqrt{2\kappa J_1^2}} \right) \right] \tag{23}$$

and

$$\mathcal{P}_z(g) = \frac{\left[ \operatorname{Im}(z - g^{-1}) \right]^{-1/2}}{\sqrt{2\pi \operatorname{Im}\langle G \rangle} J_1^2 |g|^4} v \left( \frac{\operatorname{Im}(z - g^{-1})}{J_1^2 \operatorname{Im}\langle G \rangle} \right)$$
$$\times \exp \left[ -\frac{\operatorname{Im}\langle G \rangle}{2 \operatorname{Im}(z - g^{-1})} \left( \operatorname{Re}(z - g^{-1}) - \frac{\operatorname{Im}(z - g^{-1})}{J_1^2 \operatorname{Im}\langle G \rangle} (J_0 + J_1^2 \operatorname{Re}\langle G \rangle) \right)^2 \right], \tag{24}$$

where $\langle G \rangle$ is the high-connectivity limit of the average resolvent on the cavity graph. The quantity $\langle G \rangle$ fulfills the self-consistent equation

$$\langle G \rangle = \frac{i\pi}{\sqrt{2\pi J_1^2}} \int_0^\infty d\kappa \, \kappa^{1/2} \, \nu(\kappa) \, w\left( \frac{\kappa J_0 + \kappa J_1^2 \langle G \rangle - z}{\sqrt{2\kappa J_1^2}} \right). \tag{25}$$

The function $\nu(\kappa)$ appearing in Eqs. (23-25) is the $c \to \infty$ limit of the empirical distribution of rescaled degrees,

$$\nu(\kappa) = \lim_{c \to \infty} \sum_{k=0}^\infty p_k \, \delta\left( \kappa - \frac{k}{c} \right), \tag{26}$$

while $w(\xi)$ is the Faddeeva function, defined as

$$w(\xi) = \text{erfc}(-i\xi)e^{-\xi^2}. \tag{27}$$

The properties of the Faddeeva and the complementary error function **erfc** can be found in [50].

Equations (23-25) comprise our main analytical results for the resolvent statistics of the Laplacian matrix on networks. We conclude from these equations that the high-connectivity limit of the distribution of the resolvent of $L$ is not universal, but explicitly depends on the choice of the distribution $\nu(\kappa)$ of rescaled degrees. In contrast, Eqs. (23-25) exhibit a universal behaviour with respect to the fluctuations of the coupling strengths, since they depend on the distribution $p_J$ only through its first two moments. Once we specify $\nu(\kappa)$ by means of the degree distribution $p_k$, the solutions of the self-consistent equation for $\langle G \rangle$ fully determine the spectral density $\rho_\epsilon(\lambda)$ and the distribution of the resolvent $\mathcal{P}_z(g)$ in the limit $c \to \infty$. Moreover, by integrating Eq. (24) over $\text{Re}\,g$, we can compute the distribution of the LDoS, which characterizes the spatial fluctuations of the eigenvector components.

## 4 Results for heterogeneous degrees

In this section we present numerical results for the spectral and localization properties of the Laplacian matrix on heterogeneous networks. In order to solve Eqs. (23-25), we must specify the distribution $\nu(\kappa)$ of rescaled degrees $\kappa_i = k_i/c$. Here we assume that $\kappa_i$ follows a gamma distribution

$$\nu_\gamma(\kappa) = \frac{\gamma^\gamma \kappa^{\gamma-1} e^{-\gamma\kappa}}{\Gamma(\gamma)}, \tag{28}$$

with $\gamma > 0$. The distribution $\nu_\gamma(\kappa)$ has average one and variance $1/\gamma$, providing the ideal setting for a systematic exploration of the role of degree heterogeneities by varying a single parameter. In the limit $\gamma \to \infty$, the variance goes to zero, the distribution of rescaled degrees converges to $\nu_\infty(\kappa) = \delta(\kappa - 1)$, and the network topology becomes homogeneous. In the opposite limit $\gamma \to 0$, the variance of $\nu_\gamma(\kappa)$ diverges and the network becomes strongly heterogeneous. Therefore, $\gamma$ directly controls the strength of degree fluctuations, allowing us to continuously interpolate between the homogeneous and strongly heterogeneous limits. As shown in previous work [34,51], Eq. (28) is obtained from Eq. (26) when the original degree distribution $p_k$ follows a negative binomial distribution with mean $c$ and variance $c + c^2/\gamma$.

Before we present our main findings, let us introduce a useful form of the Laplacian matrix that is valid in the limit $c \to \infty$ and makes it easier to obtain numerical diagonalization results. When $1 \ll c \ll N$, the diagonal elements in Eq. (2) are statistically independent from each other, as a consequence of the local tree-like structure of the network. In addition, each

diagonal element $L_{ii}$ is a sum of a large number of independent random variables that converges to a Gaussian variate with mean $J_0 \kappa_i$ and standard deviation $J_1 \sqrt{\kappa_i}$, with $\kappa_i$ sampled from $\nu(\kappa)$. The off-diagonal elements $L_{ij}$ ($i \neq j$) make up the weighted adjacency matrix of the network. The asymptotic forms of this adjacency matrix for $J_0 = 0$ and $J_1 = 0$, in the limit $c \to \infty$, have been put forward in references [34] and [51], respectively. By combining all these particular cases, we obtain the following asymptotic expression for the Laplacian

$$L_{ij} = \delta_{ij} \left( J_0 \kappa_i + J_1 \sqrt{\kappa_i} g_i \right) - (1 - \delta_{ij}) \left( \frac{J_0}{N} \kappa_i \kappa_j + \frac{J_1}{\sqrt{N}} \sqrt{\kappa_i \kappa_j} g_{ij} \right), \qquad (29)$$

where $g_i$ and $g_{ij} = g_{ji}$ are i.i.d. random variables drawn from a normal distribution of zero mean and unit variance. Equation (29) does not exactly fulfill the constraint $\sum_{j=1}^{N} L_{ij} = 0$ but does so *on average*, because the central limit theorem has been applied to the diagonal part of $L$. However, this is not a problem, as we are not particularly interested in the single eigenvalue $\lambda_1 = 0$ and its uniform eigenvector. Given this, the above equation provides a practical way to generate the Laplacian of a highly-connected network with an arbitrary degree distribution. When compared to conventional algorithms for sampling graphs from the configuration model with prescribed degrees [31], the use of Eq. (29) is more straightforward and computationally more efficient.

In figure 1 we compare numerical diagonalization results for the Laplacian matrix generated from Eq. (29) with those obtained by sampling random graphs from the configuration model. We assume that the rescaled degrees $\kappa_1, \ldots, \kappa_N$ follow the gamma distribution of Eq. (28), while the degrees $k_1, \ldots, k_N$ in the configuration model are sampled from a negative binomial distribution [34] with mean $c = \sqrt{N}$ and variance $c + c^2/\gamma$. Figure 1 shows that the spectral density obtained from the configuration model converges to the results derived from Eq. (29) as $N$ increases. Hence, Eq. (29) correctly reproduces the asymptotic form of $L$ in the regime where both $N$ and $c$ are infinitely large, but the ratio $c/N$ goes to zero. We point out that this high-connectivity limit is fundamentally different to the so-called dense limit, where the ratio $c/N$ remains finite as $N \to \infty$. All numerical diagonalization results presented below are obtained using Eq. (29).

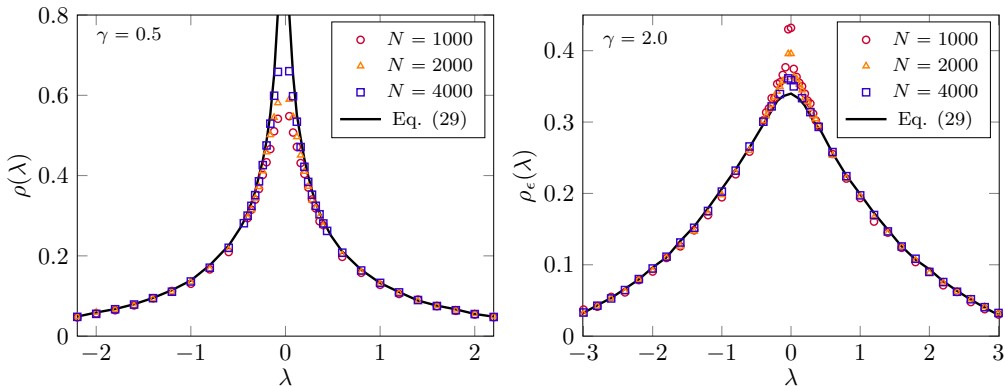

Figure 1: Numerical diagonalization results for the spectral density of the Laplacian matrix on networks. The solid black lines are obtained from Eq. (29) for $N = 2000$, with rescaled degrees $\kappa_1, \ldots, \kappa_N$ sampled from a gamma distribution with variance $1/\gamma$ (see Eq. (28)). The different symbols are obtained from random Laplacians generated according to the configuration model, where the degrees $k_1, \ldots, k_N$ are drawm from a negative binomial distribution with mean $c = \sqrt{N}$ and variance $c + c^2/\gamma$. The scaled mean and variance of $p_J$ are, respectively, $J_0 = 0$ and $J_1 = 1$.

## 4.1 Homogeneous coupling strengths ($J_1 = 0$)

We begin by setting $J_1 = 0$ and studying the spectral and localization properties of the Laplacian when there are no fluctuations in the coupling strengths. This particular regime might seem trivial at first glance. However, as we will show below, the spectral properties and the eigenvector statistics display a rich behaviour. In addition, considering the case of homogeneous coupling strengths is relevant because, for certain commonly used distributions $p_J$, such as a bimodal distribution supported in $\{A/c, B/c\}$ (where $A$ and $B$ are constants), the variance of $p_J$ is of $\mathcal{O}(1/c^2)$ and the statistics of the Laplacian resolvent are described by the solutions of Eqs. (23-25) with $J_1 = 0$. In what follows, we assume that $J_0 = 1$, since the effects of $J_0 \neq 1$ reduce to a trivial shift of the eigenvalues.

### 4.1.1 Spectral density and local density of states

By taking the limit $J_1 \to 0$ in Eqs. (23-25), we obtain expressions for the spectral density

$$\rho_\epsilon(\lambda) = \frac{1}{\pi} \, \text{Im} \int_0^\infty dx \, \frac{\nu(x)}{z-x} \tag{30}$$

and for the joint distribution of the diagonal elements of the resolvent

$$\mathcal{P}_z(g) = \frac{1}{|g|^4} \nu\big(\text{Re}(z-g^{-1})\big) \theta\big(\text{Re}(z-g^{-1})\big) \delta(\text{Im}(z-g^{-1})). \tag{31}$$

In the limit $\epsilon \to 0^+$, the Sokhotski-Plemelj identity allows one to rewrite Eq. (30) as follows

$$\rho(\lambda) = \nu(\lambda). \tag{32}$$

Therefore, the functional form of the spectral density is determined by the rescaled degree distribution $\nu(\kappa)$. Defining $y \equiv \text{Im} \, g$ as before and integrating Eq. (31) over $\text{Re} \, g$, we find an analytical result for the distribution of the LDoS,

$$\mathcal{P}_z(y) = \frac{1}{2} \sqrt{\frac{\epsilon}{y^3(1-\epsilon y)}} \left[ \nu\left(\lambda - \sqrt{\frac{\epsilon}{y}}\sqrt{1-\epsilon y}\right) \theta\left(\lambda - \sqrt{\frac{\epsilon}{y}}\sqrt{1-\epsilon y}\right) \right.$$
$$\left. + \nu\left(\lambda + \sqrt{\frac{\epsilon}{y}}\sqrt{1-\epsilon y}\right) \right] 1_{(0,1/\epsilon)}(y), \tag{33}$$

where $\theta(y)$ is the Heaviside step function and $1_{\mathcal{A}}(y)$ denotes the indicator function, i.e., $1_{\mathcal{A}}(y) = 1$ if $y \in \mathcal{A}$ while $1_{\mathcal{A}}(y) = 0$ otherwise. Note that $\mathcal{P}_z(y)$ is supported on the finite interval $(0, 1/\epsilon)$.

An alternative way of obtaining Eq. (32) is by noting that the Laplacian matrix for $J_1 = 0$ (see Eq. (29)) has the form of a diagonal matrix with entries $\tilde{\kappa}_i = \kappa_i(1 + \kappa_i/N)$, plus a rank one perturbation. From this fact, and assuming that $\tilde{\kappa}_1, \ldots, \tilde{\kappa}_N$ are arranged in ascending order, it follows [12, 52] that each of the $N-1$ intervals $(\tilde{\kappa}_i, \tilde{\kappa}_{i+1})$ contains exactly one eigenvalue $\lambda_\mu$ of $L$ (the remaining eigenvalue lies below $\tilde{\kappa}_1$). This property, known as interleaving or interlacing of eigenvalues [53], implies that for $N \to \infty$ the spectral density of the Laplacian is given by Eq. (32).

Equations (32) and (33) determine the spectral properties of the Laplacian on networks with $J_1 = 0$ and an arbitrary distribution $\nu(\kappa)$. Let us derive explicit results for $\rho(\lambda)$ when the rescaled degrees follow a gamma distribution (See Eq. (28)). Figure 2 compares Eq. (28) with numerical diagonalization results derived from an ensemble of Laplacian random matrices with $N \gg 1$ and different values of $\gamma$. The agreement with the theoretical prediction is excellent. The spectral density follows a power-law, $\rho(\lambda) \sim \lambda^{\gamma-1}$, up to $\lambda = \mathcal{O}(1/\gamma)$;

beyond this it decays exponentially fast. In particular, for strong degree fluctuations one has a divergence in the spectral density at the lower spectral edge $\lambda = 0$, similar to the behaviour of $\rho(\lambda)$ in the case of the adjacency matrix [33, 34]. Indeed, by taking the limit $\kappa \to 0$ in Eq. (28), one can see that $\rho(\lambda)$ diverges as a power-law when $0 < \gamma < 1$. For $\gamma = 1$ and $\gamma > 1$, on the other hand, we obtain $\rho(0) = 1$ and $\rho(0) = 0$, respectively.

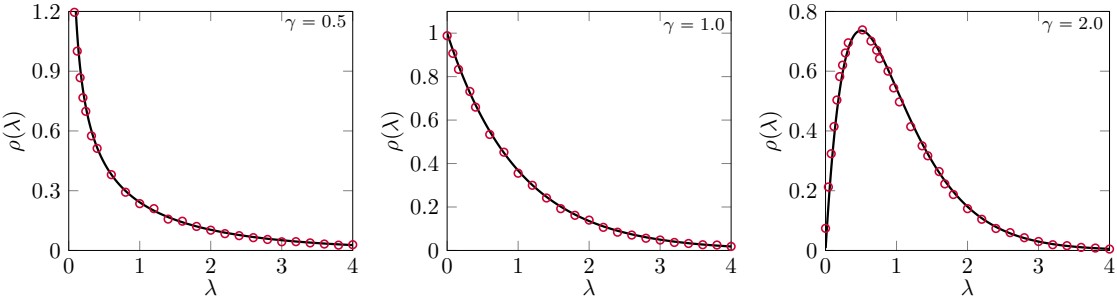

Figure 2: Spectral density of the Laplacian on highly connected random graphs characterized by a gamma distribution of rescaled degrees (see Eq. (28)) with variance $1/\gamma$. The scaled mean and variance of $p_J$ are, respectively, one and zero. The solid black lines correspond to Eq. (28). The red circles represent diagonalization results obtained from an ensemble with $100$ random Laplacians of dimension $N = 2000$ generated from Eq. (29).

We next turn our attention to the distribution $\mathcal{P}_z(y)$ of the LDoS, which encodes the spatial fluctuations of the eigenvector components. Figure 3 depicts the function $\mathcal{P}_z(y)$ calculated from Eq. (33) for the gamma distribution $\nu_\gamma(\kappa)$ of rescaled degrees, both for $\lambda = 0$ and for $\lambda > 0$. In both cases the distribution $\mathcal{P}_z(y)$ has a pronounced maximum at very small values of $y$, while for large $y$ it decays as a power-law until the upper cutoff $1/\epsilon$ is reached. The LDoS $\rho_i(z)$ is thus extremely small in a large number of nodes and very large in a tiny portion of the network, which reflects the enormous fluctuations in the eigenvector components and their localized nature [54]. The singular character of the limit $\lim_{\epsilon \to 0^+} \mathcal{P}_z(y)$ is another typical property of the localized phase [54, 55]. As a particular feature of the regime $\lambda > 0$, we note that $\mathcal{P}_z(y)$ diverges at $y_* = \epsilon/(\epsilon^2 + \lambda^2)$ as $y \to y_*^+$, for $\gamma < 1$, which results from the functional form of the gamma distribution $\nu_\gamma$.

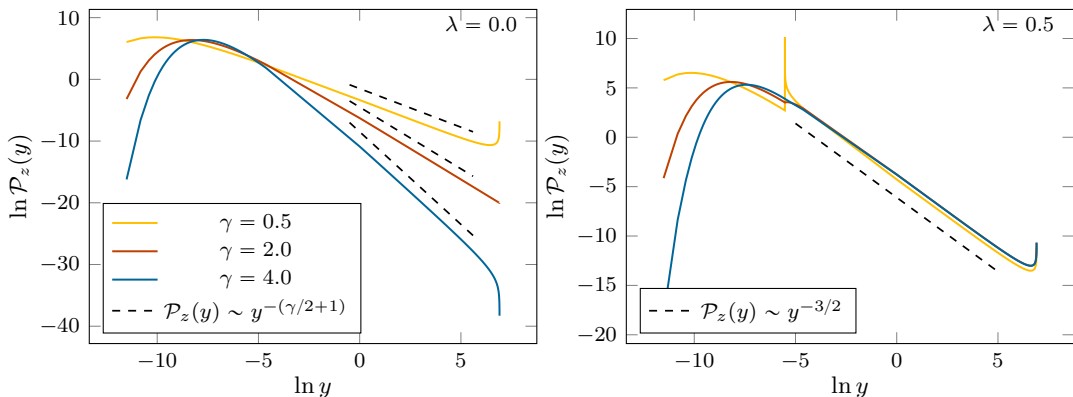

Figure 3: The distribution $\mathcal{P}_z(y)$ of the LDoS at $z = \lambda - i\epsilon$ for the Laplacian matrix on highly connected random graphs, where the rescaled degrees follow a gamma distribution with variance $1/\gamma$ (see Eq. (28)). The scaled mean and variance of $p_J$ are, respectively, one and zero. The dashed lines are the power-law functions of Eqs. (35) and (37). The coloured solid lines are obtained from Eq. (33) with $\epsilon = 10^{-3}$.

Let us extract the right (large $y$) tail of $\mathcal{P}_z(y)$, since its functional form is particularly relevant for computing the multifractal exponents of the eigenvectors. For $\lambda = 0$, Eq. (33) reduces to

$$\mathcal{P}_z(y) = \frac{1}{2}\sqrt{\frac{\epsilon}{y^3(1-\epsilon y)}}\, \nu\left(\sqrt{\frac{\epsilon}{y}}\sqrt{1-\epsilon y}\right) \mathbf{1}_{(0,1/\epsilon)}(y), \tag{34}$$

which shows that the right tail of $\mathcal{P}_z(y)$ is determined by the functional form of $\nu(\kappa)$. When the rescaled degrees follow $\nu_\gamma(\kappa)$ (see Eq. (28)), Eq. (34) yields the asymptotic expression

$$\mathcal{P}_z(y) \propto \epsilon^{\gamma/2} y^{-\gamma/2-1} \ (\epsilon \ll y \ll 1/\epsilon), \tag{35}$$

valid as $\epsilon \to 0^+$. The $\epsilon$-dependent prefactor in Eq. (35) implies that, in the regime of $\gamma < 1$, $\int_\epsilon^{1/\epsilon} y\mathcal{P}_z(y)dy \sim \epsilon^{\gamma-1}$, which is consistent with the behaviour of $\rho(\lambda)$ at $\lambda = 0$ (see Eq. (28)).

Turning next to $\lambda > 0$, with again $\epsilon \ll y \ll 1/\epsilon$, the $\epsilon$-dependent contribution in the argument of $\nu$ in Eq. (33) is vanishingly small, leading to the approximate form

$$\mathcal{P}_z(y) \simeq \sqrt{\frac{\epsilon}{y^3}}\, \nu(\lambda)\, \mathbf{1}_{(0,1/\epsilon)}(y), \tag{36}$$

which shows that the exponent governing the decay of $\mathcal{P}_z(y)$ is independent of $\nu(\kappa)$, that is

$$\mathcal{P}_z(y) \propto \epsilon^{1/2} y^{-3/2} \ (\epsilon \ll y \ll 1/\epsilon). \tag{37}$$

The integral $\int_\epsilon^{1/\epsilon} y\mathcal{P}_z(y)dy$ is now always dominated by the large-$y$ power law tail, giving together with the $\epsilon$-dependent prefactor a result of $\mathcal{O}(1)$. This agrees with the fact that $\rho(\lambda)$ is always finite for $\lambda \neq 0$. The power-law tail of $\mathcal{P}_z(y)$ results from strong spatial fluctuations of the eigenvector components throughout the network. Figure 3 illustrates the power-law decay of $\mathcal{P}_z(y)$ for the gamma distribution of rescaled degrees, confirming Eqs. (35) and (37).

### 4.1.2 Singularity spectrum and multifractal exponents

In this subsection, we determine the singularity spectrum and the multifractal exponents of the eigenvectors within the bulk of the spectral density, i.e., for an arbitrary $\lambda > 0$. Since

the distribution of the LDoS characterizes the fluctuations of the squared eigenvector amplitudes around a given $\lambda$, it is sensible to assume that the right tail of the distribution $P_\psi(x)$ of $x_i = N|\psi_i|^2$ behaves according to the LDoS distribution. Thus, based on Eq. (37), we assume that $P_\psi(x)$ decays as

$$P_\psi(x) \sim b(N)x^{-3/2} \quad (x > x_{\min}(N)), \tag{38}$$

for $x$ above some characteristic value $x_{\min}(N)$. The scaling of $x_{\min}(N)$ and of the prefactor $b(N)$ with respect to $N$ follows from the normalization conditions

$$\int_{x_{\min}}^{N} P_\psi(x)dx \simeq 1 \tag{39}$$

and

$$\int_{x_{\min}}^{N} x P_\psi(x)dx \simeq 1. \tag{40}$$

By substituting Eq. (38) into the above constraints, we find $b(N) \sim N^{-1/2}$ and $x_{\min}(N) \sim N^{-1}$. Inserting then the resulting expression $P_\psi(x) \sim N^{-1/2}x^{-3/2}$ into Eq. (14), we obtain the singularity spectrum

$$f(\alpha) = \frac{1}{2}\alpha \quad \text{for} \quad \alpha \in [0, 2), \tag{41}$$

where the lower and upper ends of the range of $\alpha$ are determined by the conditions $f(\alpha) = 0$ and $f(\alpha) = 1$, respectively [43].

Moving to small values of $x$, we find that the distribution $P_\psi(x)$ behaves as

$$P_\psi(x) \sim N^{\gamma/2}x^{\gamma/2-1}. \tag{42}$$

In appendix B, we demonstrate the validity of Eqs. (38) and (42) by a direct computation of the eigenvectors of $L$ as defined in Eq. (29). By substituting Eq. (42) into Eq. (14), we derive the singularity spectrum in the range of $\alpha$ corresponding to small $x$,

$$f(\alpha) = \frac{\gamma}{2}(2-\alpha) + 1 \quad \text{for} \quad \alpha \in (2, 2 + 2/\gamma], \tag{43}$$

Summarizing Eqs. (41) and (43), the function $f(\alpha)$ has a triangular shape. This is shown in figure 4, where we present our analytic predictions for $f(\alpha)$ together with numerical diagonalization results of Eq. (29), for $\lambda > 0$ and two different values of $\gamma$ (see numerical details in Appendix C). The diagonalization results agree very well with Eqs. (41) and (43), especially for low values of $\gamma$, where the argument for the right piece of $f(\alpha)$ becomes quite accurate already for moderate $N$. Equation (41) implies that the eigenvectors for $\lambda > 0$ are localized [39], regardless of the value of $\gamma$. This behaviour is consistent with the Anderson-like tail of Eq. (37) [36] as well as with the singular behaviour of the LDoS distribution in the limit $\epsilon \to 0^+$, as discussed in the previous subsection.

We now turn our attention to the set of multifractal exponents $\tau(q)$. For $q > 0$, only the left piece of the triangular function $f(\alpha)$, given by Eq. (41), is relevant for the calculation of the Legendre transform (see Eq. (12)). The resulting expression for $\tau(q)$ reads

$$\tau(q) = \begin{cases} 2q - 1 & \text{for} \quad q \leq \frac{1}{2}, \\ 0 & \text{for} \quad q > \frac{1}{2}. \end{cases} \tag{44}$$

Consistently with the behaviour of the singularity spectrum, Eq. (44) reveals the localized nature of the eigenvectors, which can be clearly seen in the numerical diagonalization results

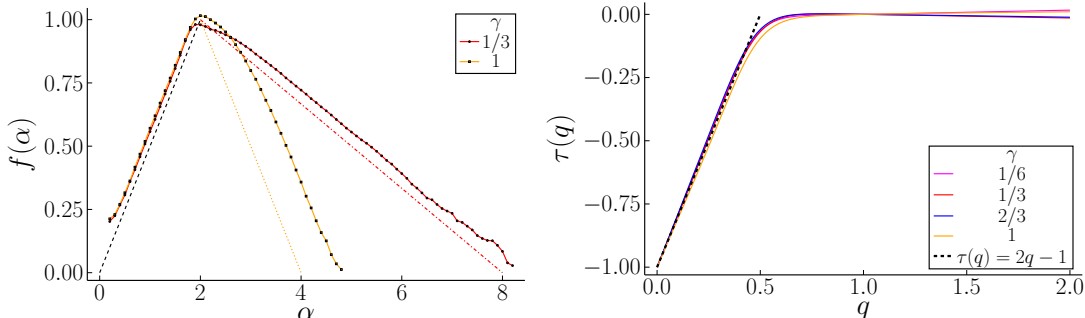

Figure 4: Singularity spectrum $f(\alpha)$ and multifractal exponents $\tau(q)$ of the eigenvectors corresponding to $\lambda = 1$ of the Laplacian matrix on highly-connected random graphs with homogeneous coupling strengths ($J_0 = 1$ and $J_1 = 0$). The parameter $\gamma$ controls the variance of the distribution of rescaled degrees (see Eq. (28)). Left panel: solid lines with markers correspond to numerical diagonalization results. Dashed line: analytical prediction of Eq. (41). Dash-dotted lines: analytical predictions from Eq. (43). Right panel: solid lines represent numerical diagonalization results, while the dashed line corresponds to Eq. (44).

shown in figure 4. We stress that $\tau(q)$ is independent of $\gamma$ and $\lambda$, as long as we choose $\lambda$ far from the spectral edge $\lambda = 0$, as specified above. Note also that this set of multifractal exponents corresponds exactly to the behavior of localized wavefunctions at the critical point of the Anderson transition on random graphs [38, 40].

## 4.2 Heterogeneous coupling strengths ($J_1 > 0$)

In this section we discuss the spectral properties and the eigenvector statistics of the Laplacian matrix on highly-connected networks that combine the effect of heterogeneous degrees and random coupling strengths.

### 4.2.1 Spectral density and local density of states

First, let us recover a previous rigorous result for the spectral density valid in the homogeneous network topology limit $\gamma \to \infty$ [28]. Substituting $\nu_\infty(\kappa) = \delta(\kappa-1)$ into Eqs. (23) and (25), we obtain the regularized spectral density

$$\rho_\epsilon(\lambda) = \frac{1}{\pi}\mathrm{Im}\langle G\rangle, \tag{45}$$

where $\langle G\rangle$ fulfills

$$\langle G\rangle = \frac{i\pi}{\sqrt{2\pi J_1^2}}\, w\left(\frac{J_0 + J_1^2\langle G\rangle - z}{\sqrt{2J_1^2}}\right). \tag{46}$$

For $J_0 = 0$ and $J_1 = 1$, $\rho_\epsilon(\lambda)$ is a symmetric distribution around $\lambda = 0$, given by the free additive convolution of the Wigner semicircle law with the normal distribution, as rigorously demonstrated in [28]. Even if Eqs. (45) and (46) have been derived here using the non-rigorous cavity method, they are exact for highly connected random graphs in the limit $N \to \infty$, allowing us to put forward a sensible conjecture that generalizes the theorem in [28] for $J_0 \neq 0$. Indeed, the above equations, combined with the decomposition of the Laplacian, Eq. (29), strongly suggest that $\rho(\lambda)$ is given by the free additive convolution of the Wigner semicircle law supported in $[-2J_1, 2J_1]$ with a Gaussian distribution of mean $J_0$ and variance

$J_1^2$. Therefore, a nonzero $J_0$ shifts the mode of $\rho(\lambda)$. Figure 5 confirms Eqs. (45) and (46) by comparing the outcome of solving these equations with numerical diagonalization results for the spectral density.

Furthermore, our approach enables us to calculate the distribution of the LDoS and study the statistics of the eigenvector components. By inserting $\nu_\infty(\kappa) = \delta(\kappa - 1)$ into Eq. (24), performing the limit $\epsilon \to 0^+$ and integrating over $\mathrm{Re}\,g$, we obtain the analytical result

$$\mathcal{P}_\lambda(y) = \frac{\exp\left(\frac{-x_+^2(y)}{2J_1^2}\right) + \exp\left(\frac{-x_-^2(y)}{2J_1^2}\right)}{\sqrt{2\pi J_1^2 y^3 \left(y_e - y\right)}} \mathbf{1}_{(0,y_e)}(y), \tag{47}$$

where the upper end $y_e$ of the support of $\mathcal{P}_\lambda(y)$ is given by

$$y_e = \frac{1}{J_1^2 \pi \rho(\lambda)}. \tag{48}$$

The functions $x_\pm(y)$ in Eq. (47) are defined as

$$x_\pm(y) = \lambda - J_0 - J_1 \mathrm{Re}\langle G\rangle \pm \frac{1}{y_e}\sqrt{\frac{y_e}{y} - 1}. \tag{49}$$

The fact that, in the limit $\epsilon \to 0^+$, the marginal $\mathcal{P}_z(y) = \int_{-\infty}^{\infty} d\mathrm{Re}\,g\, \mathcal{P}_z(\mathrm{Re}\,g, y)$ converges to a regular, $\epsilon$-independent function $\mathcal{P}_\lambda(y)$ supported on a finite interval, is a characteristic property of the extended phase [54,55]. Figure 5 exhibits the LDoS distribution obtained from Eq. (47) for different $\lambda$. As $\lambda$ increases, $\mathcal{P}_\lambda(y)$ becomes gradually more concentrated around its typical value, which shifts towards smaller values of $y$ at the same time.

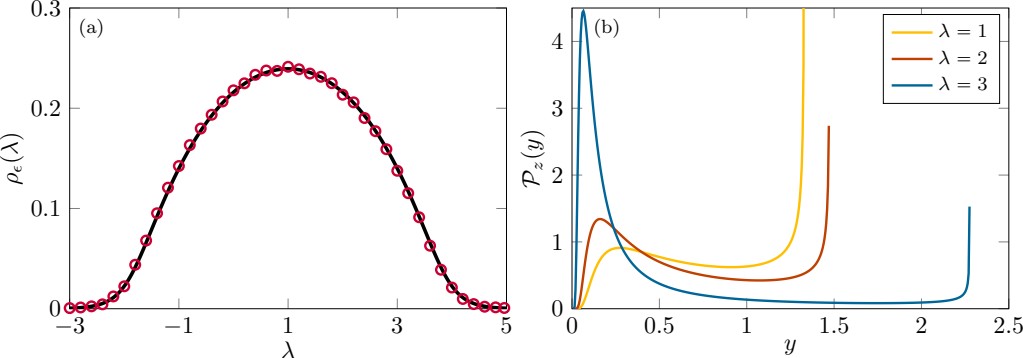

Figure 5: Spectral density $\rho_\epsilon(\lambda)$ and the distribution $\mathcal{P}_z(y)$ of the LDoS for the Laplacian matrix on highly-connected random graphs in the homogeneous regime ($\nu_\infty(\kappa) = \delta(\kappa - 1)$). The scaled first and second moments of the coupling strengths are, respectively, $J_0 = 1$ and $J_1 = 1$. In panel (a) we compare the theoretical results (solid line), obtained from the solutions of Eqs. (45) and (46) for $\epsilon = 0$, with numerical data (red circles) obtained from the diagonalization of 100 random Laplacians generated from Eq. (29) with $N = 2000$. In panel (b) we present the distribution of the LDoS, Eq. (47), for different $\lambda$.

Finally, we consider the situation where $J_1 > 0$ and the rescaled degrees are distributed according to Eq. (28). In this case, the regularized spectral density follows from

$$\rho_\epsilon(\lambda) = \frac{\gamma^\gamma}{\Gamma(\gamma)\sqrt{2\pi J_1^2}} \mathrm{Re}\left[\int_0^\infty d\kappa\, \kappa^{\gamma - 3/2} e^{-\gamma\kappa}\, w\left(\frac{\kappa J_0 + \kappa J_1^2 \langle G\rangle - z}{\sqrt{2\kappa J_1^2}}\right)\right], \tag{50}$$

where $\langle G \rangle$ solves

$$\langle G \rangle = \frac{\gamma^\gamma i \pi}{\Gamma(\gamma)\sqrt{2\pi J_1^2}} \int_0^\infty d\kappa \, \kappa^{\gamma-1/2} e^{-\gamma\kappa} \, w\left( \frac{\kappa J_0 + \kappa J_1^2 \langle G \rangle - z}{\sqrt{2\kappa J_1^2}} \right). \tag{51}$$

We calculate the integrals over $\kappa$ in the above equations using numerical methods. From a computational point of view, obtaining a numerical solution of these integrals is more efficient than performing numerical diagonalizations of high-dimensional matrices or solving Eqs. (21) and (22) using the population dynamics algorithm [26].

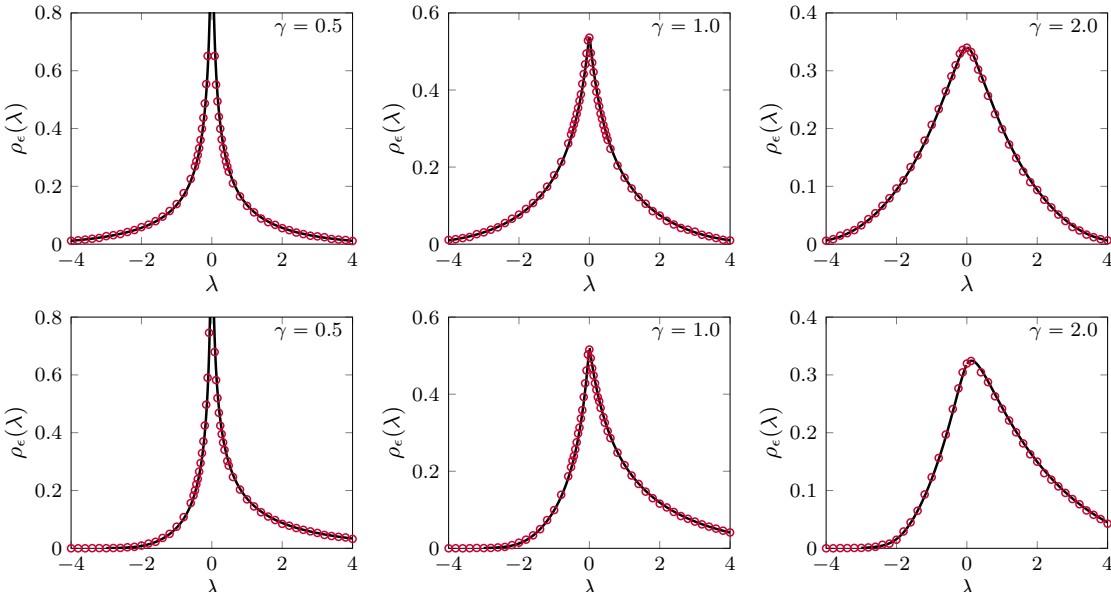

Figure 6: Spectral density of the Laplacian on highly-connected random graphs characterized by a gamma distribution of rescaled degrees with variance $1/\gamma$ (see Eq. (28)). The scaled mean and variance of $p_J$ are, respectively, $J_0$ and $J_1 = 1$. The upper panels show results for $J_0 = 0$, while in the lower row $J_0 = 1$. The solid black lines are obtained from the solutions of Eqs. (50) and (51) with $\epsilon = 10^{-3}$. The red circles represent diagonalization results obtained from an ensemble with **100** Laplacian random matrices of dimension $N = 2000$ generated from Eq. (29).

Figure 6 shows the spectral density $\rho_\epsilon(\lambda)$ obtained from the solutions of Eqs. (50) and (51), together with numerical diagonalization results of random Laplacians generated from Eq. (29). The excellent agreement between the theoretical curves and the numerical diagonalization data confirms the exactness of Eqs. (50) and (51). The upper and lower rows show results for $J_0 = 0$ and $J_0 = 1$, respectively. Comparing these, one notices that the mode of the distribution is independent of $J_0$, but a nonzero value of $J_0$ breaks the symmetry of $\rho_\epsilon(\lambda)$ around $\lambda = 0$, yielding an excess of positive (negative) eigenvalues when $J_0 > 0$ ($J_0 < 0$). For $J_0 \neq 0$, the spectral density becomes gradually more symmetric around $\lambda = 0$ as $\gamma \to 0$, since the shift in the eigenvalues caused by a finite mean $J_0$ becomes less relevant than the spread of eigenvalues induced by strong degree fluctuations. Similarly to the behaviour of the spectral density of the adjacency matrix [34], $\rho_\epsilon(\lambda)$ diverges at $\lambda = 0$ for $\gamma \leq 1/2$. We point out that for $J_1 \neq 0$ the Laplacian matrix has both positive and negative eigenvalues, irrespective of whether the network is homogeneous or heterogeneous.

As our final set of results for the spectral properties, we analyse the LDoS distribution $\mathcal{P}_z(y)$ for $J_1 > 0$ and $\nu(\kappa)$ given by the gamma distribution. We compute $\mathcal{P}_z(y)$ by numerically

integrating Eq. (24) over the real part $\mathbf{Re}\,g$. Figure 7 illustrates the effect of degree fluctuations on the distribution $\mathcal{P}_z(y)$ for $\lambda = 0$ and $\lambda \neq 0$. In either case, $\mathcal{P}_z(y)$ has an unbounded support for any finite $\gamma$, which is an important difference with respect to the LDoS distribution of the adjacency matrix, since in the latter case the support is unbounded only at $\lambda = 0$ [34]. As $\gamma$ increases, $\mathcal{P}_z(y)$ slowly converges to the distribution of the LDoS for homogeneous networks (see figure 5). For $\lambda > 0$, the function $\mathcal{P}_z(y)$ diverges at $y = 0$ when $\gamma < 1$, owing to the functional form of the distribution $\nu_\gamma(\kappa)$ of rescaled degrees.

The insets in figure 7 illustrate the large-$y$ behaviour of $\mathcal{P}_z(y)$ for both $\lambda = 0$ and $\lambda \neq 0$. The distribution $\mathcal{P}_z(y)$ exhibits an exponential tail for $\lambda > 0$, implying that the first moment of $\mathcal{P}_z(y)$ and, consequently, the spectral density, is finite. For $\lambda = 0$, the LDoS distribution decays for large $y$ as

$$\mathcal{P}_z(y) \propto \frac{1}{y^{\gamma+3/2}}, \tag{52}$$

with an exponent $\gamma + 3/2$ controlled by the strength of degree heterogeneities. Due to the power-law tail of Eq. (52), the first moment of $\mathcal{P}_z(y)$ diverges when $\gamma \leq 1/2$, which explains the singular behaviour of the spectral density at $\lambda = 0$ (see figure 6). In what follows, we will use Eq. (52) to make an *ansatz* for $P_\psi(x)$ and extract the singularity spectrum $f(\alpha)$.

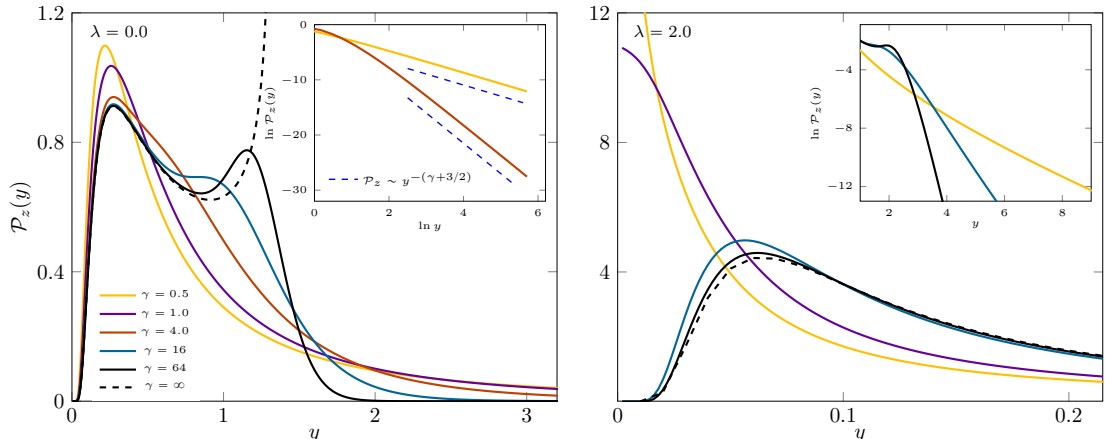

Figure 7: The distribution $\mathcal{P}_z(y)$ of the LDoS at $z = \lambda - i\epsilon$ for the Laplacian matrix on highly connected random graphs, where the rescaled degrees follow a gamma distribution with variance $1/\gamma$ (see Eq. (28)). The scaled mean and variance of $p_J$ are, respectively, $J_0 = 0$ and $J_1 = 1$. The solid lines for different $\gamma$ are obtained by marginalizing Eq. (24) with $\epsilon = 0$, while the dashed lines in the main panels represent the homogeneous limit of $\mathcal{P}_z(y)$ (see Eq. (47)). The insets exhibit the tails of $\mathcal{P}_z(y)$ for large $y$.

### 4.2.2 Singularity spectrum and multifractal exponents

In this subsection, we focus on the eigenvector statistics around $\lambda = 0$, at which the spectral density diverges for $\gamma \leq 1/2$. In contrast to the case of homogeneous coupling strengths, where the eigenvalues are non-negative and $\rho(\lambda)$ can diverge only at the spectral edge $\lambda = 0$, here the divergence occurs within the bulk of the spectrum. This is a similar scenario as the one analyzed in references [34, 35] for the weighted adjacency matrix of the same class of networks. We follow reference [35] and assume that $\mathcal{P}_z(y)$ provides a good estimate for the distribution $P_\psi(x)$ of the eigenvector squared amplitudes $x_i = N|\psi_i|^2$ corresponding to $\lambda = 0$. Thus, we take as our *ansatz* for $P_\psi(x)$, above some scale $x_{\min}(N)$ and up to the upper bound

$N$, the form from Eq. (52),

$$P_\psi(x) = b(N)x^{-(\gamma+3/2)} \quad (x > x_{\min}(N)), \tag{53}$$

where $b(N)$ is an $N$-dependent prefactor. In Fig. 8, we show that the exponent in Eq. (53) is consistent with the numerical results for $P_\psi(x)$ obtained from numerical diagonalizations of random Laplacians with finite $N$. In addition, the data in figure 8 suggest that $P_\psi(x)$ behaves as

$$P_\psi(x) \sim x^{-1/2} \tag{54}$$

for small $x$. This scaling holds independently of $\gamma$.

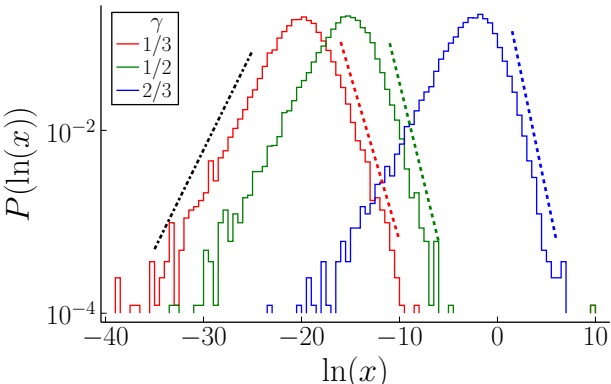

Figure 8: Distribution $P_\psi(\ln x)$ of the logarithm of squared eigenvector amplitudes $x_i = N|\psi_i|^2$ for the Laplacian matrix on highly-connected hetereogenous random graphs ($J_0 = 0$ and $J_1 = 1$), for different values of the parameter $\gamma$ that controls the rescaled degree distribution (see Eq. (28)). The figure depicts $P_\psi(\ln x)$ for the eigenvector corresponding to the closest eigenvalue to $\lambda = 0$ (the vertical axis is in logarithmic scale). These results are obtained by diagonalizing random Laplacian matrices generated from Eq. (29) with $N = 2^{14}$. Dashed lines: scaling according to $P(\ln x) \sim e^{-(\gamma+1/2)\ln x}$ (see Eq. (53)). Dash-dotted line: scaling according to $P(\ln x) \sim e^{\ln x/2}$ (see Eq. (54)).

Let us now compute the singularity spectrum and the multifractal exponents, using Eqs. (14) and (12). The normalization conditions of Eqs. (39) and (40) determine how the prefactor $b(N)$ and the lower end $x_{\min}(N)$ of the power-law range of Eq. (53) scale with $N$. For $\gamma < 1/2$, one finds $b(N) \sim N^{\gamma-1/2}$ and $x_{\min}(N) \sim N^{\frac{\gamma-1/2}{\gamma+1/2}}$; for $\gamma > 1/2$, both $b(N)$ and $x_{\min}(N)$ are independent of $N$. By inserting Eq. (53) into Eq. (14), together with the scaling of $b(N)$, we can estimate the singularity spectrum for $N \to \infty$. For $\gamma < 1/2$, we obtain

$$f(\alpha) = (\gamma + 1/2)\alpha \quad \text{for} \quad 0 \le \alpha \le \frac{1}{\gamma + 1/2}, \tag{55}$$

while the Legendre transform of the above expression gives the corresponding multifractal exponents (see Eq. (12))

$$\tau(q) = \begin{cases} \dfrac{q}{\gamma + 1/2} - 1 & \text{for} \quad q < \gamma + 1/2, \\ 0 & \text{for} \quad q > \gamma + 1/2. \end{cases} \tag{56}$$

In the regime $\gamma > 1/2$, an analogous calculation yields the singularity spectrum

$$f(\alpha) = \alpha\left(\gamma + \frac{1}{2}\right) + \left(\frac{1}{2} - \gamma\right) \quad \text{for} \quad \frac{\gamma - 1/2}{\gamma + 1/2} \le \alpha \le 1, \tag{57}$$

with the corresponding multifractal exponents

$$\tau(q) = \begin{cases} q - 1 & \text{for} \quad q \le \gamma + 1/2, \\ q\dfrac{\gamma - 1/2}{\gamma + 1/2} & \text{for} \quad q > \gamma + 1/2. \end{cases} \tag{58}$$

The intervals of $\alpha$ in Eqs. (55) and (57) are determined from the conditions $f(\alpha) = 0$ and $f(\alpha) = 1$ as above [43]. Regarding the behaviour of $f(\alpha)$ for large $\alpha$, some comments are in order. By virtue of Eq. (14) the behaviour of $P_\psi(x)$ for small $x$, given by Eq. (54), implies that the singularity spectrum behaves as $f(\alpha) \approx \text{const} - \alpha/2$ for $\alpha$ larger than some characteristic value. This additional piece of the singularity spectrum does not have any effect on the function $\tau(q)$ for $q > 0$, but it might be relevant for classifying the eigenvectors into universality classes. Taken together, these results suggest that $f(\alpha)$ again has a triangular shape, analogous to what we found in section 4.2.2 for the homogeneous case.

Equations (55) and (56) imply that the eigenvectors for $\gamma < 1/2$ are localized with strong multifractal behaviour [40]. Indeed, the eigenvectors for single instances and finite connectivity turn out to be localized on a finite number of nodes that are not spatially correlated. In this regard, the eigenvectors exhibit similar features as those of the weighted adjacency matrices analyzed in reference [35]. Equations (57) and (58), in contrast, imply that the eigenvectors for $\gamma > 1/2$ are extended and non-ergodic [46]. Notice that, in the limit $\gamma \to \infty$, the eigenvectors become fully ergodic since $f(\alpha)$ shrinks to a single point with value $f(1) = 1$, while $\tau(q) = q - 1$ for all $q$. To summarize, then, $\gamma = 1/2$ defines the critical point that characterizes a localization-delocalization transition induced by fluctuations in the network degrees.

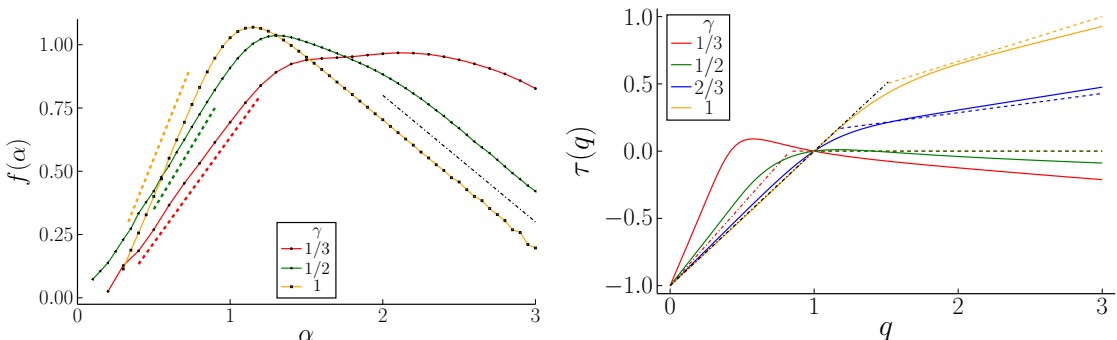

Figure 9: Singularity spectrum $f(\alpha)$ and multifractal exponents $\tau(q)$ of the eigenvectors corresponding to $\lambda = 0$ of the Laplacian matrix on highly-connected random graphs with heterogeneous coupling strengths ($J_0 = 0$ and $J_1 = 1$). The parameter $\gamma$ controls the variance of the rescaled degree distribution (see Eq. (28)). Left panel: solid lines with markers correspond to numerical diagonalization data, dashed lines indicate the slope of $f(\alpha) = (\gamma + 1/2)\alpha$ (see Eqs. (55) and (57)), and the dash-dotted line shows the slope of $f(\alpha) \approx \text{const} - \alpha/2$. Right panel: solid lines are numerical diagonalization results, while the dashed lines correspond to Eqs. (56) and (58).

We end this section by comparing our theoretical predictions with numerical data. Figure 9 presents numerical diagonalization results for the singularity spectrum and the mutlifractal exponents for different values of $\gamma$ (we refer to Appendix C for numerical details regarding

the computation of these functions). Equations (55) and (57) predict that $f(\alpha)$ has a slope $\gamma + 1/2$ for small $\alpha$, while Eq. (54) yields a slope of $-1/2$ for large $\alpha$. These theoretical findings are consistent with the numerical diagonalization results for $f(\alpha)$ presented in figure 9. As mentioned above, we expect that $f(\alpha)$ approaches a triangular shape as $N \to \infty$. However, as can be seen in figure 9, the numerical results show significant deviations from a triangle-like function for low $\gamma$ (strong degree fluctuations). We interpret this as an indication of strong finite-size effects. Such effects are also visible in the multifractal exponents, where we observe, for low $\gamma$, a mismatch between the theoretical prediction (Eqs. (56) and (58)) and the numerical diagonalization results. On the other hand, the theoretical results for $\tau(q)$ in the regime $\gamma \geq 1/2$ (see Eq. (58)) exhibit a very good agreement with the numerical data.

## 5  Summary and conclusions

In this study we have performed a thorough analysis of the spectral and localization properties of the Laplacian matrix on highly-connected networks with an arbitrary degree distribution and random coupling strengths. We have shown that, as the average degree $c$ grows to infinity, the distribution of the diagonal resolvent entries converges to a relatively simple closed form expression, which depends on the full distribution of rescaled degrees and the first two moments of the coupling strengths. This implies that the high-connectivity limit of the spectral properties of the Laplacian on networks is universal with respect to the statistics of the coupling strengths. However, this universality breaks down in the presence of strong degree fluctuations, akin to what is observed for adjacency matrices [33]. The analytical expression for the resolvent distribution serves as the basis for systematically studying how the heterogeneous structure of random graphs impacts the spectral density, the distribution of the local density of states (LDoS), the singularity spectrum and the multifractal exponents, leading to a comprehensive picture of the spectral and localization properties of the Laplacian on heterogeneous networks. We have focused on highly-connected networks with a gamma distribution of rescaled degrees, as in this case the strength of degree fluctuations is solely controlled by the variance $1/\gamma$ of the rescaled degree distribution. The network becomes homogeneous for $\gamma \to \infty$, while the strong heterogeneous limit is reached as $\gamma \to 0$.

When the variance of the coupling strengths is nonzero (heterogeneous couplings), the spectral density diverges within the bulk of the spectrum, provided $\gamma \leq 1/2$. This singularity is a consequence of the large fluctuations of the LDoS, whose distribution exhibits a power-law decay governed by a $\gamma$-dependent exponent. These results for the Laplacian spectral properties are qualitatively similar to those for the adjacency matrix [34]. By using the power-law tail of the LDoS distribution as an input, we have computed the singularity spectrum and the multifractal exponents, which characterize the spatial fluctuations of the squared eigenvector amplitudes. Our findings show that the emergence of the singularity in the eigenvalue distribution is accompanied by a delocalization-localization transition of the corresponding eigenvectors. For $\gamma > 1/2$, the spatial fluctuations of the eigenvectors with eigenvalues around zero are characteristic of non-ergodic extended states, while for $\gamma < 1/2$ these eigenvectors become localized, exhibiting strong multifractal behaviour. The existence of a non-ergodic extended phase has been demonstrated in various random-matrix models for the diffusion of a single quantum particle on random structures, including the Rosenzweig-Porter ensemble [46, 56], the Cayley tree [57] and the sparse Erdös-Rényi ensemble [58, 59]. In contrast to these settings, the localization transition in the present model is driven by strong degree fluctuations.

When the variance of the coupling strengths is zero (homogeneous couplings), the Laplacian of highly-connected networks has only non-negative eigenvalues, and the functional form of the spectral density is given by the rescaled degree distribution. In this case, the spectral

density only diverges at the lower spectral edge. By focusing on the eigenvector statistics within the bulk of the spectrum, we have shown that the LDoS distribution displays a singular behaviour as the regularizer $\epsilon$ tends to zero. Furthermore, this function decays as a power-law with exponent $3/2$, which is also the value characterizing the power-law tail of the LDoS in the whole localized phase of the Anderson model on a Bethe lattice (including the critical point) [36, 55]. These findings imply that all eigenvectors within the bulk of the Laplacian spectrum are localized, regardless of the variance $1/\gamma$ of the rescaled degrees. The behaviour of both the multifractal exponents and the singularity spectrum is also consistent with the localized nature of the wavefunctions at the critical point of the Anderson model on random graphs [39, 40, 60]. This is interesting because the localization length of models with spatial structure diverges precisely at the critical point, which is consistent with the absence of any notion of distance in the current network model. Thus, the localized states in the present model are not related to the spatial structure of the graph, since the Laplacian effectively becomes a fully-connected matrix in the limit $c \to \infty$ (see Eq. (29)). Instead, the eigenvectors for large but finite-sized networks localize onto nodes with degrees of $\mathcal{O}(1)$, which is typical of the statistical localization mechanism introduced in [35].

We point out that our results for homogeneous coupling strengths match exactly those for the master operator of the mean-field Bouchaud trap model [12, 61], provided one identifies the temperature in that model with the parameter $\gamma$ here. Indeed, both random-matrix models are defined (at least effectively, in the case studied here) in terms of a fully-connected network and all eigenvectors are localized, regardless of the value of the control parameter. The main difference between these two models lies in the source of disorder. While in the current model the disorder stems from the network degrees, in the Bouchaud trap model the local energies are quenched random variables. This unexpected connection between the two models suggest that statistical localization is a widespread phenomenon, which can be triggered by distinct physical mechanisms.

Based on the central limit theorem and previous rigorous results [28, 62], we have put forward an asymptotic form of the Laplacian matrix, Eq. (29), valid in the limit $c \to \infty$. This equation provides an efficient way to diagonalise finite-size instances of the Laplacian without having to sample graphs from the configuration model. Besides this practical advantage, we point out that the off-diagonal elements of Eq. (29) define the adjacency matrix of an effective fully-connected interaction matrix that incorporates the effect of degree fluctuations in the high-connectivity limit. This fully-connected network model has recently attracted significant interest as it leads to exactly solvable models in various contexts [51, 63–65]. It would be interesting to study the tight-binding Anderson model with on-site random potentials (diagonal disorder) and hopping energies given by the off-diagonal entries of Eq. (29). This model would enable one to characterize the transition between statistical localization (absence of diagonal disorder) [35] and Anderson-like localized states (strong diagonal disorder).

Finally, we have derived numerical diagonalization results of finite Laplacians that, in general, exhibit a very good agreement with our theoretical findings. The only exception is the regime of strong degree fluctuations (small $\gamma$) and heterogeneous coupling strengths. In this case, even though the data are consistent overall with the analytical equations for the singularity spectrum and the multifractal exponents, the numerical results in figure 9 show significant deviations from the theory, which require further analysis. One could complement our results and achieve a more complete characterization of the delocalization-localization transition in the present model by computing other spectral observables, such as the level-spacing distribution, the level compressibility and the overlap correlation function [42].

# Acknowledgements

**Funding information**    JDS acknowledges a fellowship from CNPq/Brazil. FLM thanks CNPq/Brazil for financial support. Simulations for the eigenvector statistics were run on the GoeGrid cluster at the University of Göttingen, which is supported by the Deutsche Forschungsgemeinschaft (project IDs 436382789; 493420525).

# A   The high-connectivity limit of the resolvent equations

In previous works [33, 34], we have introduced a method for deriving analytical expressions for the spectral observables of the adjacency matrix of random graphs with arbitrary degree distributions in the high connectivity limit. In this section, we extend this approach to the Laplacian matrix.

We start by defining the joint distribution $\mathcal{W}_z(s)$ of the real and imaginary parts of

$$S(z) \equiv \sum_{\ell=1}^{k} \frac{J_\ell}{1 - J_\ell g_\ell(z)}, \tag{A.1}$$

which consists of a sum of independent and identically distributed random variables. The degree $k$ is sampled from the discrete distribution $p_k$, the coupling strengths $\{J_\ell\}$ follow the distribution $p_J$, while the cavity resolvents $\{g_\ell(z)\}$ are drawn from $\mathcal{Q}_z(g)$. The random variable $S(z)$ is analogous to the self-energy that appears in the solution of tight-binding models for the diffusion of an electron [54]. The distribution $\mathcal{W}_z(s)$ of $S(z)$ is determined from

$$\mathcal{W}_z(s) = \sum_{k=0}^{\infty} p_k \int_{\mathbb{H}^+} \Big[ \prod_{\ell=1}^{k} dg_\ell \, \mathcal{Q}_z(g_\ell) \Big] \int_{\mathbb{R}} \Big[ \prod_{\ell=1}^{k} dJ_\ell p_J(J_\ell) \Big] \delta\Big( s - \sum_{\ell=1}^{k} \frac{J_\ell}{1 - J_\ell g_\ell} \Big). \tag{A.2}$$

The argument of the Dirac-$\delta$ in Eq. (21) means that

$$G(z) \overset{\mathrm{d}}{=} \frac{1}{z - S(z)}, \tag{A.3}$$

which leads to the following relation between $\mathcal{P}_z(g)$ and $\mathcal{W}_z(s)$

$$\mathcal{P}_z(g) = \int_{\mathbb{H}^+} ds \, \mathcal{W}_z(s) \delta\Big( g - \frac{1}{z - s} \Big), \tag{A.4}$$

with $ds = d\mathrm{Re}\, s \, d\mathrm{Im}\, s$. Equation (A.4) implies that all moments of $\mathcal{P}_z(g)$ are determined by $\mathcal{W}_z(s)$. For instance, the regularized spectral density $\rho_\epsilon(\lambda)$, written in terms of the distribution $\mathcal{W}_z(s)$, reads

$$\rho_\epsilon(\lambda) = \frac{1}{\pi} \mathrm{Im} \Big( \int_{\mathbb{H}^+} ds \, \frac{\mathcal{W}_z(s)}{z - s} \Big). \tag{A.5}$$

In a similar way, one can introduce the distribution $\mathcal{V}_z(s)$ of the self-energy on the cavity graph,

$$\mathcal{V}_z(s) = \sum_{k=1}^{\infty} \frac{k}{c} p_k \int_{\mathbb{H}^+} \Big[ \prod_{\ell=1}^{k-1} dg_\ell \, \mathcal{Q}_z(g_\ell) \Big] \int_{\mathbb{R}} \Big[ \prod_{\ell=1}^{k-1} dJ_\ell p_J(J_\ell) \Big] \delta\Big( s - \sum_{\ell=1}^{k-1} \frac{J_\ell}{1 - J_\ell g_\ell} \Big), \tag{A.6}$$

which allows us to rewrite $\mathcal{Q}_z(g)$ as follows (see Eq. (22))

$$\mathcal{Q}_z(g) = \int_{\mathbb{H}^+} ds \, \mathcal{V}_z(s) \delta\Big( g - \frac{1}{z - s} \Big). \tag{A.7}$$

Our goal is to determine the form of the distributions $\mathcal{W}_z(s)$ and $\mathcal{V}_z(s)$ in the limit $c \to \infty$. Since $\mathcal{W}_z(s)$ and $\mathcal{V}_z(s)$ are distributions of sums of independent and identically distributed random variables, it is natural to work with the corresponding characteristic functions. Let $\varphi_{\mathcal{W}}(p,t)$ and $\varphi_{\mathcal{V}}(p,t)$ be the characteristic functions of $\mathcal{W}_z(s)$ and $\mathcal{V}_z(s)$, respectively, defined as the Fourier transforms

$$\varphi_{\mathcal{W}}(p,t) = \int_{\mathbb{H}^+} ds\, \mathcal{W}_z(s) \exp(-ip\,\mathrm{Re}\,s - it\,\mathrm{Im}\,s), \tag{A.8}$$

$$\varphi_{\mathcal{V}}(p,t) = \int_{\mathbb{H}^+} ds\, \mathcal{V}_z(s) \exp(-ip\,\mathrm{Re}\,s - it\,\mathrm{Im}\,s). \tag{A.9}$$

The substitution of Eqs. (A.2) and (A.6) into Eqs. (A.8) and (A.9) produces

$$\varphi_{\mathcal{W}}(p,t) = \sum_{k=0}^{\infty} p_k \exp[k\mathcal{S}_c(p,t)] \tag{A.10}$$

and

$$\varphi_{\mathcal{V}}(p,t) = \sum_{k=1}^{\infty} \frac{k}{c} p_k \exp[(k-1)\mathcal{S}_c(p,t)], \tag{A.11}$$

with

$$\mathcal{S}_c(p,t) = \ln\left[\iint_{\mathbb{H}^+} dg\, \mathcal{Q}_z(g) \int_{-\infty}^{\infty} dJ\, p_J(J) \exp\left(\frac{-ip(J - J^2\mathrm{Re}\,g) - it(J^2\mathrm{Im}\,g)}{(1 - J\mathrm{Re}\,g)^2 + (J\mathrm{Im}\,g)^2}\right)\right]. \tag{A.12}$$

The next step is to expand Eq. (A.12) for $c \gg 1$.

In order to proceed further, we remind the reader that the mean and variance of $p_J$ are, respectively, given by $J_0/c$ and $J_1^2/c$, while higher moments are proportional to $1/c^\beta$ ($\beta > 1$). Therefore, the leading term in Eq. (A.12) for $c \gg 1$ is given by

$$\mathcal{S}_c(p,t) = -\frac{ip}{c}J_0 - \frac{ip}{c}\mathrm{Re}\langle G\rangle J_1^2 - \frac{it}{c}\mathrm{Im}\langle G\rangle J_1^2 - \frac{p^2}{2c}J_1^2 + \mathcal{O}(1/c^2), \tag{A.13}$$

where we have assumed that the average resolvent on the cavity graph,

$$\langle G\rangle = \int_{\mathbb{H}^+} ds\, \frac{\mathcal{V}_z(g)}{z - s}, \tag{A.14}$$

converges to a well-defined limit as $c \to \infty$. Substituting Eq. (A.13) into Eqs. (A.10) and (A.11), and using the definition of the distribution of rescaled degrees (Eq. (26)) we obtain

$$\varphi_{\mathcal{W}}(p,t) = \int_0^{\infty} d\kappa\, \nu(\kappa) \exp\left[\kappa\left(-ipJ_0 - ipJ_1^2\mathrm{Re}\langle G\rangle - it J_1^2\mathrm{Im}\langle G\rangle - p^2 J_1^2/2\right)\right] \tag{A.15}$$

and

$$\varphi_{\mathcal{V}}(p,t) = \int_0^{\infty} d\kappa\, \kappa\, \nu(\kappa) \exp\left[\kappa\left(-ipJ_0 - ipJ_1^2\mathrm{Re}\langle G\rangle - it J_1^2\mathrm{Im}\langle G\rangle - p^2 J_1^2/2\right)\right]. \tag{A.16}$$

The inverse Fourier transforms of Eqs. (A.15) and (A.16) yield, respectively,

$$\mathcal{W}_z(s) = \frac{1}{\sqrt{2\pi J_1^2}} \int_0^{\infty} d\kappa\, \kappa^{-1/2} \nu(\kappa) \exp\left[-\frac{1}{2J_1^2\kappa}\left(\mathrm{Re}\,s - \kappa\left(J_0 + J_1^2\mathrm{Re}\langle G\rangle\right)\right)^2\right]$$
$$\times \delta\left(\mathrm{Im}\,s - \kappa J_1^2\mathrm{Im}\langle G\rangle\right) \tag{A.17}$$

and

$$
\mathcal{V}_z(s) = \frac{1}{\sqrt{2\pi J_1^2}} \int_0^\infty d\kappa \, \kappa^{1/2} \, v(\kappa) \exp\left[ -\frac{1}{2J_1^2 \kappa} \left( \mathrm{Re}\, s - \kappa \left( J_0 + J_1^2 \mathrm{Re}\langle G \rangle \right) \right)^2 \right]
$$
$$
\times \, \delta \left( \mathrm{Im}\, s - \kappa J_1^2 \mathrm{Im}\langle G \rangle \right). \tag{A.18}
$$

The above two equations form the central result of this appendix, as they provide exact equations for the spectral observables. By substituting Eq. (A.17) into Eq. (A.5) and integrating over $s \in \mathbb{H}^+$, we recover Eq. (23) for the spectral density. Similarly, the self-consistent Eq. (25) for the average resolvent $\langle G \rangle$ on the cavity graph is obtained by substituting Eq. (A.18) into (A.14) and then solving the integral over $s \in \mathbb{H}^+$.

Finally, let us derive Eq. (24) for the distribution $\mathcal{P}_z(g)$ of the diagonal elements of the resolvent. Integrating over $\kappa$ in Eq. (A.17), we get

$$
\mathcal{W}_z(s) = \frac{1}{\sqrt{2\pi} J_1^3 \mathrm{Im}\langle G \rangle} \overline{v}\left( \frac{\mathrm{Im}\, s}{J_1^2 \mathrm{Im}\langle G \rangle} \right) \exp\left[ -\frac{\mathrm{Im}\langle G \rangle}{2\mathrm{Im}\, s} \left( \mathrm{Re}\, s - \frac{\mathrm{Im}\, s}{J_1^2 \mathrm{Im}\langle G \rangle} \left( J_0 + J_1^2 \mathrm{Re}\langle G \rangle \right) \right)^2 \right],
$$
$$\tag{A.19}$$

where we defined $\overline{v}(\kappa) \equiv k^{-1/2} v(\kappa)$. The resolvent $G(z)$ is related to the self-energy $S(z)$ through Eq. (A.3). Since we know the analytical form (A.19) of the joint distribution $\mathcal{W}_z(s)$ of $S(z)$, Eq. (24) for the distribution $\mathcal{P}_z(g)$ of $G(z)$ is derived by making the two-dimensional change of variables dictated by Eq. (A.3).

# B   The distribution of the eigenvector squared amplitudes for homogeneous coupling strengths

In this appendix, we show how to derive Eqs. (38) and (42), which characterize the distribution $P_\psi(x)$ of the eigenvector squared amplitudes for large and small $x$. From the asymptotic form of the Laplacian for $J_1 = 0$ (see Eq. (29)), the eigenvalue equation (3) can be written as

$$
\kappa_i \psi_{\mu,i} - \frac{1}{N} \kappa_i \sum_{j=1(\neq i)}^N \kappa_j \psi_{\mu,j} = \lambda_\mu \psi_{\mu,i}, \tag{B.1}
$$

where we have considered $J_0 = 1$, as was done in Section 4.1. By converting the above sum into an unconstrained sum, the eigenvalue equation becomes

$$
\psi_{\mu,i} \left( \tilde{\kappa}_i - \lambda_\mu \right) = \kappa_i A_\mu, \tag{B.2}
$$

where we have defined

$$
A_\mu = \frac{1}{N} \sum_{j=1}^N \kappa_j \psi_{\mu,j} \tag{B.3}
$$

and

$$
\tilde{\kappa}_i = \kappa_i (1 + \kappa_i / N). \tag{B.4}
$$

Thus, we can write $\psi_{\mu,i}$ as

$$
\psi_{\mu,i} = \frac{A_\mu \kappa_i}{\tilde{\kappa}_i - \lambda_\mu}. \tag{B.5}
$$

Multiplying the above equation by $\kappa_i$ and then averaging over $i$ must give back $A_\mu$, leading to the eigenvalue condition

$$1 = \frac{1}{N} \sum_i \frac{\kappa_i^2}{\tilde{\kappa}_i - \lambda_\mu}, \tag{B.6}$$

from which the interleaving property referred to in the main text follows.

Once an eigenvalue $\lambda$ is found, the eigenvector components obey (see Eq. (B.5))

$$\psi_i = \frac{A\kappa_i}{\kappa_i - \lambda}, \tag{B.7}$$

where we have approximated $\tilde{\kappa}_i \simeq \kappa_i$ for large $N$. For the sake of simplicity, we have also dropped the eigenvalue index $\mu$. Our purpose here is to understand how the distribution $P_\psi(x)$ of $x_i = N|\psi_i|^2$ scales for large and small values of $x_i$. Focussing on eigenvalues away from the spectral edge, i.e. when $\lambda = \mathcal{O}(1)$, one has to distinguish between three regimes according to whether $\kappa_i$ is (i) much smaller, (ii) comparable to or (iii) much larger than $\lambda$. To be specific, let us assume that the sequence $\kappa_1, \ldots, \kappa_N$ is arranged in ascending order, and let $j$ be such that $\lambda$ lies between $\kappa_j$ and $\kappa_{j+1}$. Thus, the regimes (i) and (iii) correspond to $|i - j| = O(N)$, which yield

$$\psi_i \approx \begin{cases} -A\kappa_i/\lambda & \text{for} \quad \kappa_i \ll \lambda, \\ A & \text{for} \quad \kappa_i \gg \lambda. \end{cases} \tag{B.8}$$

In the regime (ii), where $|i - j| \ll N$, we can approximate

$$\kappa_i - \lambda \approx \kappa_i - \kappa_j \approx \frac{i - j}{N\nu(\kappa_j)}. \tag{B.9}$$

Equations (B.8) and (B.9) are valid as long as $|i - j| \gg 1$, when there is a large number of other $\kappa_k$ lying between $\kappa_i$ and $\kappa_j$. In particular, the second relation in Eq. (B.9) comes from the fact that the interval $[\kappa_i, \kappa_j]$ typically contains $N(\kappa_j - \kappa_i)\nu(\kappa_j)$ rescaled degrees out of all the $N$ values of $\kappa_k$. Thus, in regime (ii) we have

$$\psi_i \approx \frac{AN\nu(\kappa_j)}{i - j}. \tag{B.10}$$

The constant $A$ can be determined from the normalization condition $\sum_{i=1}^N \psi_i^2 = 1$. From Eq. (B.10), one gets a contribution to the sum of

$$\sum_{i:|i-j|\ll N} \frac{A^2 N^2 \nu^2(\kappa_j)}{|i - j|^2} \sim A^2 N^2. \tag{B.11}$$

On the other hand, regimes (i) and (iii) from Eq. (B.8) give $\psi_i = \mathcal{O}(A)$ and hence a contribution of $O(NA^2)$ to the normalization $\sum_{i=1}^N \psi_i^2$. This is negligible in comparison to the one coming from regime (ii). We therefore conclude that $A \sim N^{-1}$, which allows us to obtain $x_i$ in the three different regimes:

$$x_i = N|\psi_i|^2 \sim \begin{cases} N^{-1}\kappa_i^2 & \text{for} \quad \kappa_i \ll \lambda, \\ N|i - j|^{-2} & \text{for} \quad 1 \ll |i - j| \ll N, \\ N^{-1} & \text{for} \quad \kappa_i \gg \lambda, \end{cases} \tag{B.12}$$

where we have omitted all factors of order unity. The largest $x_i$ come from the second regime, in which $1 \ll |i - j| \ll N$. In this case, the probability that $x$ is larger than some characteristic

value $\bar{x}$ is obtained by counting what fraction of the index values obey $|i-j| < (N/\bar{x})^{1/2}$. This fraction scales as $N^{-1}(N/x)^{1/2}$, which yields the behaviour of the probability density $P_\psi(x)$ for large $x$ as

$$P_\psi(x) \sim -\frac{d}{dx}N^{-1}(N/x)^{1/2} \sim N^{-1/2}x^{-3/2}. \tag{B.13}$$

The smallest $x_i$, on the other hand, come from the regime (i), where $\kappa_i \sim \sqrt{Nx_i}$. In this regime we obtain

$$P_\psi(x) \sim \nu\left(\sqrt{Nx}\right)\sqrt{N/x} \tag{B.14}$$

Assuming now that $Nx \ll 1$, we can simplify this using the behaviour of the gamma distribution for small argument (see Eq. (28)), which leads to

$$P_\psi(x) \sim N^{\gamma/2}x^{\gamma/2-1}. \tag{B.15}$$

## C  Numerical methods for the eigenvector statistics

### C.1  Singularity spectrum

For a given eigenvector $\vec{\psi}$ of size $N$ around the eigenvalue $\lambda$, we compute the set of exponents $\{\alpha_i\}_{i=1}^N$ via the expression

$$\alpha_i = -\frac{\ln(|\psi_i|^2)}{\ln(N)}. \tag{C.1}$$

We collect these exponents for a band of **50** eigenvectors around $\lambda$ obtained from an ensemble with **10** random instances of the Laplacian matrix generated according to Eq. (29) and $N = 2^{15}$. The histogram of the exponents estimates the distribution $\Omega(\alpha)$ (Eq. (9)). Thus, by using Eq. (10), we estimate the singularity spectrum as

$$f(\alpha) \approx \frac{\ln\Omega(\alpha)}{\ln N} + 1. \tag{C.2}$$

Since Eq. (10) is an asymptotic scaling relation for large $N$, the resulting estimates are affected by finite-size corrections.

### C.2  Multifractal exponents

We determine $\tau(q)$ by estimating for each $q$ the typical value of the $q$-th moment of $|\psi_i|^2$ across a band of **50** eigenvectors around $\lambda$. In practice, we consider a grid of values of $q$ with a step size $\Delta q = 0.02$, network sizes $N \in \{2^{14}, 2^{15}\}$, and we generate $2^{21}/N$ instances of the Laplacian matrix for each $N$ according to Eq. (29). For a given eigenvector, we compute $I_q(N)$ using Eq. (7), and then calculate $I_q^{\text{typ}}(N) = e^{\langle \ln I_q(N)\rangle}$ with the average taken over the ensemble of eigenvectors and instances. Finally, by virtue of the scaling (8), we extract $\tau(q)$ by estimating $I_q^{\text{typ}}$ for two different values of $N$, generically denoted as $N_1$ and $N_2$. The formula

$$\tau(q) = -\frac{\left[\ln I_q^{\text{typ}}(N_2) - \ln I_q^{\text{typ}}(N_1)\right]}{\ln N_2 - \ln N_1} \tag{C.3}$$

yields an estimate for $\tau(q)$. In our case, we consider $N_2 = 2N_1 = 2^{15}$, so that Eq. (C.3) reduces to

$$\tau(q) = -\frac{\ln I_q^{\text{typ}}(N_2) - \ln I_q^{\text{typ}}(N_1)}{\ln 2}. \tag{C.4}$$

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
