# Peer review of "Spectral properties, localization transition and multifractal eigenvectors of the Laplacian on heterogeneous networks"

_SciPost Physics_

## Round 1 · Referee Report · Anonymous (Referee 1) · 2024-10-12

Strengths

  1. Interesting and very well supported results,
  2. Clarity of the exposition.

Weaknesses

Most of the technical breakthroughs had already been presented in previous publications.

Report

In this manuscript, the authors consider the Laplacian matrix on highly connected networks with a gamma distribution of the degrees, and with random weights. The spectral and localization properties of the model are encoded in the solution of the resolvent (cavity) distributional equations, which is here extracted analytically in the large-connectivity limit. This allows to analyze in great depth not only the spectral density of the model, but also the distribution of the LDoS, which in turn gives access to the localization properties of the eigenvectors. The singularity spectrum and multifractal exponents are then evaluated as a function of the (inverse) variance $\gamma$ of the rescaled degrees, separately for the case of constant or random coupling strengths. While in the former case the eigenvectors in the bulk of the spectrum are always localized, the latter case exhibits a transition from non-ergodic delocalized states to localized eigenvectors when the degree fluctuations become sufficiently strong (which happens at $\gamma=1/2$). This is accompanied by the divergence of the spectral density in the bulk of the spectrum.

I found the paper very well written and pedagogical in its exposition, and I think that it provides a thorough and comprehensive analysis of the model under consideration.
I would not hesitate to recommend its publication in SciPost Physics, but in fact I’m not yet entirely convinced of the originality of these results. Indeed, the analytic solution of the cavity equations in the high-connectivity limit was first presented in Ref. [34] (by authors 1 and 4 of the present manuscript), although for the case of the adjacency matrix (instead of the Laplacian). Such solution allowed for the analysis of the spectral density and IPR on heterogeneous random graphs. This study did not include the characterization of the multifractal exponents, which was later addressed in Ref. [35] (by authors 2 and 3), where the mechanism of “statistical localization” was also unveiled and described. The authors themselves state all of this very clearly already in the Introduction, and also elsewhere in the present manuscript.
Although I did appreciate a lot the organic and self-contained nature of this work, I'd encourage the authors to comment clearly on how it distinguishes itself from Refs. [34-35], apart from of course replacing the adjacency matrix by the Laplacian matrix. For example, in the present case the value $\gamma=1/2$ marks an interesting transition from non-ergodic delocalized to localized states: did the same happen already for the case of the adjacency matrix?

Below I add a few remarks and suggestions.

Requested changes

  1. On p. 5, there is probably an “as” missing from “the entire degree distribution acts a control parameter”.

  2. On p. 6, the sentence “the empirical spectral density in Eq. (4) is the first moment of $P_\lambda (y)$” might be promoted as a displayed equation; this would allow to refer to it later, e.g. in the discussion that follows Eq. (35).

  3. In Eq. (21) and in Appendix A, can the meaning of the integration over $H^+$ be spelled out?

  4. On p. 9, “Eqs. (23-25) exhibit a universal behaviour with respect to the fluctuations of the coupling strengths, since they depend on the distribution $p_J$ only through its first two moments.” However, earlier on p. 5 it was assumed “that higher-order moments of $p_J$ are proportional to $1/c^\beta$, with $\beta>1$”. Aren’t these two points related? Is it surprising that only the first two moments of the distribution intervene into Eqs. (23-25)?

  5. Right panel of Fig. 1: why in this case $\rho_\epsilon$ was plotted instead of $\rho$, that appears instead on the left panel? Can the authors comment on the slight discrepancy observed in the middle of the spectrum?

  6. In Fig. 4, the yellow line is not really “dash-dotted”: does it still correspond to Eq. (43)? Why is the agreement with the numerical data getting worse at $\gamma=1$, whereas in Fig. 9(left) the opposite seems to occur?

  7. In Eq. (46), it may be worth recalling that $w$ is the Faddeeva function introduced earlier in Eq. (27). Moreover, it may be worth mentioning that $w$ is in fact the Cauchy-Stieltjes transform associated to a Gaussian distribution, which is after all the reason why Eq. (46) suggests a free additive convolution.

  8. In Sec. 4.2.2, the strong degree fluctuations for $\gamma= 1/3$ unfortunately affect negatively the numerical measurements. Still, a way to further support the correctness of the singularity spectrum depicted in Fig. 9(left) may be to check if $f(\alpha)$ satisfies the very general symmetry proposed in [PRL 97, 046803 (2006)].

Recommendation

Ask for minor revision

  • validity: high
  • significance: high
  • originality: low
  • clarity: top
  • formatting: perfect
  • grammar: perfect

Author:  Fernando Metz  on 2024-11-25  [id 4988]

(in reply to Report 1 on 2024-10-12)

Referee:In this manuscript, the authors consider the Laplacian matrix on highly connected networks with a gamma distribution
of the degrees, and with random weights. The spectral and localization properties of the model are encoded in the solution
of the resolvent (cavity) distributional equations, which is here extracted analytically in the large-connectivity limit.
This allows to analyze in great depth not only the spectral density of the model, but also the distribution of the LDoS, which
in turn gives access to the localization properties of the eigenvectors. The singularity spectrum and multifractal exponents
are then evaluated as a function of the (inverse) variance $\gamma$ of the rescaled degrees, separately for the case of
constant or random coupling strengths. While in the former case the eigenvectors in the bulk of the spectrum are always
localized, the latter case exhibits a transition from non-ergodic delocalized states to localized eigenvectors when the
degree fluctuations become sufficiently strong (which happens at $\gamma=1/2$). This is accompanied by the divergence
of the spectral density in the bulk of the spectrum.

Referee:I found the paper very well written and pedagogical in its exposition, and I think that it provides a thorough and
comprehensive analysis of the model under consideration.
I would not hesitate to recommend its publication in SciPost Physics, but in fact I’m not yet entirely
convinced of the originality of these results. Indeed, the analytic solution of the cavity equations in the
high-connectivity limit was first presented in Ref. [34] (by authors 1 and 4 of the present manuscript), although
for the case of the adjacency matrix (instead of the Laplacian). Such solution allowed for the analysis of the
spectral density and IPR on heterogeneous random graphs. This study did not include the characterization of the
multifractal exponents, which was later addressed in Ref. [35] (by authors 2 and 3), where the mechanism
of “statistical localization” was also unveiled and described. The authors themselves state all of this very
clearly already in the Introduction, and also elsewhere in the present manuscript.
Although I did appreciate a lot the organic and self-contained nature of this work, I'd encourage the authors
to comment clearly on how it distinguishes itself from Refs. [34-35], apart from of course replacing the
adjacency matrix by the Laplacian matrix. For example, in the present case the value $\gamma=1/2$ marks an
interesting transition from non-ergodic delocalized to localized states: did the same happen already for the case of the adjacency matrix?

We thank the referee for the careful reading of our manuscript and for the opportunity to clarify the novelty of our work. Indeed,
the referee's question at the end of this comment already contains the nucleus of the answer: the work reported in our manuscript shows
that the spectral physics of the adjacency matrix and the Laplacian are quite distinct for heterogeneous networks, in contrast to
homogeneous networks where degree fluctuations are negligible so that Laplacian and adjacency matrix differ only trivially, by a
multiple of the identity matrix. This key physical result is novel and is exemplified by the fact that $\gamma=1/2$ marks a qualitative
transition in the spectral properties of the Laplacian, while for the adjacency matrix case a transition occurs only at $\gamma=1$.

As regards the methodological aspects, the
referee is correct in observing that -- for adjacency matrices -- the distribution of the LDoS has been analytically computed in reference [34]. In the subsequent work [35], the power-law
tail of this LDoS distribution is used as an input to determine the multifractal exponents $\tau(q)$ and the singularity spectrum $f(\alpha)$ in the regime $\gamma < 1$. In
addition, reference [35] introduces
the notion of statistical localization, which accounts for the behaviour of the eigenvectors for small $\gamma$. These results presented in [34,35] thus provide a detailed
characterization of the multifractal localized phase observed for $\gamma < 1$. However, these works do not include an
explicit computation of $\tau(q)$ and $f(\alpha)$ for $\gamma >1$, and while a localization-delocalization transition is expected at $\gamma=1$, a comprehensive study
of this transition is not presented in references [34,35].

In the present work, we compute $\tau(q)$ and $f(\alpha)$ for the Laplacian matrix on high-connectivity networks and we
explicitly show, for the first time, that there is a transition between
a delocalized non-ergodic phase and a multifractal localized phase as the variance $1/\gamma$ of the rescaled degree distribution increases.
The comprehensive analysis of this transition, which is absent from references [34,35], is a further important aspect of the originality of our work.
The analytic tools developed
in the present work will be useful to study the spectral and localization properties of other random-matrix models with diagonal disorder, such as the Anderson model on a
highly-connected random graph with on-site disorder.

Following the referee's suggestion, we have included an extra appendix in the paper which summarizes previous results
for $\tau(q)$ and $f(\alpha)$ for the adjacency matrix and thus makes the contrast to our new results for the Laplacian clearer.

Referee:Below I add a few remarks and suggestions.

Referee: On p. 5, there is probably an “as” missing from “the entire degree distribution acts a control parameter”.

We thank the referee for this observation. The text has been corrected.

Referee: On p. 6, the sentence ``the empirical spectral density in Eq. (4) is the first moment of $P_\lambda(y)$'' might be promoted as a displayed
equation; this would allow to refer to it later, e.g. in the discussion that follows Eq. (35).

We thank the referee for the suggestion. We have added a new equation on page 6 that explicitly shows the relation
between $\rho(\lambda)$ and $P_\lambda(y)$.

Referee: In Eq. (21) and in Appendix A, can the meaning of the integration over $\mathbb H^+$ be spelled out?

We have implemented the referee’s suggestion and reminded the reader of the meaning of the symbol $\mathbb H^+$ after Eq. (21) and in Appendix A.

Referee: On p. 9, ``Eqs. (23-25) exhibit a universal behaviour with respect to the fluctuations of the coupling
strengths, since they depend on the distribution $p_J$ only through its first two moments.'' However, earlier
on p. 5 it was assumed ``that higher-order moments of $p_J$ are proportional to $1/c^\beta$, with $\beta>1$''. Aren’t
these two points related? Is it surprising that only the first two moments of the distribution intervene into Eqs. (23-25)?

The referee is right in noting that it is not surprising that only the first two moments of $p_J$ appear in Eqs. (23-25), given
that higher-order moments are proportional to $1/c^{\beta}$, with $\beta>1.$ However, our intention with the
aforementioned sentence and the paragraph after Eq. (27) is
to highlight that Eqs. (23-25) do depend on the specific form of the rescaled degree distribution $\nu(\kappa)$, while they do not depend on the specific
distribution $p_J$, but only on its first two moments. We slightly changed the text after Eq. (27) to clarify this point.

Referee: Right panel of Fig. 1: why in this case $\rho_\epsilon$ was plotted instead of $\rho$, that appears instead on the left panel? Can the authors comment on
the slight discrepancy observed in the middle of the spectrum?

Regarding the first question, the fact we plotted $\rho_{\epsilon}$ instead of $\rho$ was a typo, which has been corrected. The slight discrepancy observed in
the middle of the spectrum is due to finite-size effects resulting from finite values of the mean degree $c=\sqrt{N}$. To make this point clear, in the attached figure we compare the
distribution $\nu_{\gamma,c}(\kappa) = c \, p_{\kappa c}$ ($\kappa=0,\frac{1}{c},\frac{2}{c},\dots$) of rescaled degrees for finite $c$ with the gamma distribution $\nu_{\gamma}(\kappa)$
of rescaled degrees, valid in the limit $c \rightarrow \infty$. Clearly, as $c$ increases, $\nu_{\gamma,c}(\kappa)$ converges to $\nu_{\gamma}(\kappa)$, but there is
a discrepancy for small values of $\kappa$. The fraction of nodes with small rescaled degrees is larger for smaller values of $c$, resulting in the
discrepancy of $\rho(\lambda)$ around $\lambda=0$ (small eigenvalues are associated with low-degree nodes). We do not think this effect is sufficiently
important to be discussed in the paper, since the numerical results in Figure 1, obtained from the configuration model, are clearly converging to the results derived from Eq. (30) as $N$ increases.

Referee: In Fig. 4, the yellow line is not really ``dash-dotted'': does it still correspond to Eq. (43)? Why is the agreement with the
numerical data getting worse at $\gamma=1$, whereas in Fig. 9(left) the opposite seems to occur?

We thank the referee for pointing this out. We have fixed the line style according to the caption. The yellow line still
corresponds to Eq. (43). In contrast with Figure 9, which is for heterogeneous coupling strengths at $\lambda = 0$, Figure 4
corresponds to homogeneous coupling strengths at $\lambda = 1$. The
trend in both figures is different because the underlying phenomena are different. In the homogeneous case, the eigenvectors are localized for any value of $\gamma$. In the
heterogeneous case, the eigenvectors are extended for $\gamma > 1/2$ and strong finite size effects appear for $\gamma < 1/2$, which is the scenario in which statistical localization occurs.

Referee: In Eq. (46), it may be worth recalling that $w$ is the Faddeeva function introduced earlier in Eq. (27). Moreover, it may be
worth mentioning that $w$ is in fact the Cauchy-Stieltjes transform associated to a Gaussian distribution, which is after all the reason why Eq. (46) suggests a free additive convolution.

The text below Eq. (46) has been slightly modified according to the referee's suggestions.

Referee: In Sec. 4.2.2, the strong degree fluctuations for $\gamma=1/3$ unfortunately affect negatively the numerical
measurements. Still, a way to further support the correctness of the singularity spectrum depicted in Fig. 9(left) may be to
check if $f(\alpha)$ satisfies the very general symmetry proposed in [PRL 97, 046803 (2006)].

This is an interesting point that amounts to checking that the multifractal states satisfy the symmetry relation $f(\alpha + 1) = f(1 - \alpha) + \alpha$ (see
appendix A of Kravtsov, Altshuler and Ioffe, Annals of Physics \textbf{389}, 148 (2018)).
However, a quick calculation shows that this relation is inconsistent with the result that follows from Eq. (54) of the resubmitted paper, namely, that
the slope of the right piece of the singularity
spectrum is $-1/2$, while the slope of the left piece depends on $\gamma$. Additionally, and as discussed in the reference mentioned by the referee, this symmetry
relation implies that the singularity spectrum is supported in $\alpha \in [0, 2]$, which is not the case here. Finally, we point out that, even though the symmetry relation is
satisfied by a broad class of systems, it has also been shown that it is not universally satisfied. A counterexample is given by the localized
states of the Anderson model on a Bethe lattice away from the critical point, as discussed in reference [39].

Attachment:

gamma_dist.pdf

---

## Round 1 · Referee Report · Anonymous (Referee 2) · 2024-10-14

Strengths

1- Clear, understandable and original 2- Clever use of an analytically solvable case and sophisticated cavity method techniques 3- Interesting exposition of the strongly multifractal regime around the $\lambda = 0$ states of the weighted Laplacian matrix. 4- Contrast with the more traditionally localized case and the extended case make the paper accessible to those unfamiliar with more exotic phenomenology

Weaknesses

1- A "weakly multifractal" phase for $\lambda \neq 0$ is likely to be there, but this is not really commented upon or explored. 2- What makes the results for the Laplacian distinct from those of the adjacency matrix could be more thoroughly commented upon 3- The approximation of the graph Laplacian by a weighted fully-connected graph is only tested using the average DoS. A numerical test comparing the full statistics of the LDoS (or the multifractal exponents) for actual configuration model networks and the "annealed" approximation might be a good idea.

Report

The manuscript by da Silva, Tapias, Sollich and Metz is a nice collaboration combining ideas from the authors' previous works. Tapias and Sollich recently described the strongly multifractal states around $\lambda = 0$ for the adjacency matrix of weighted configuration model networks (notably following da Silva and Metz in using the gamma distribution of rescaled degrees). Together with da Silva and Metz, the ensemble of authors have now exploited some clever new tricks, in combination with the trusty cavity method, to gain insight into the Laplacian matrix of such networks.

The work is clear, suitably pedagogical, and gives the reader a rather intuitive picture of why we ought to expect novel behaviour of the eigenstates in this model: In the case with no off-diagonal disorder, the Laplacian has only diagonal disorder (dictated by the degree distribution). Naturally, this leads to localized states, with an average DoS dictated by the degree distribution. When all nodes have the same degree but off-diagonal disorder is allowed, there is no diagonal disorder and the states are extended, with a DoS that follows a shifted version of (what I think is) Fyodorov's result. When we have both diagonal and off-diagonal disorder, one therefore expects that there could be an interesting mix between the two extremes.

A thorough analysis is provided, showing the behaviour of the multifractal exponents, the singularity spectrum and the local and average DoS in selected cases. Particular attention is paid to the $\lambda = 0$, where one finds a novel localization transition (as a function of the degree heterogeneity) between localized multifractal states and non-ergodic delocalized states with weak multifractality. This also coincides with a transition from singular to finite DoS at $\lambda = 0$.

I would recommend publication after minor revision. I have only the following comments and queries.

1- Have the authors also considered the case of the normalized Laplacian (where the rows are divided by the degree of the node in question, such that the diagonal elements are all equal to unity -- see e.g. Ref [29] of the manuscript)? In many applications, it is the normalized version that is used, but I can imagine that the phenomenology might differ quite a lot to that of the version used here. Certainly, the DoS is different (one obtains the Wigner semi-circle in the dense limit), and in the fully localized phase with $J_1= 0$ the states would become degenerate. I of course do not recommend that the same thorough analysis be carried out for the normalized version, but a comment on this could be elucidating for readers.

2- As is also discussed in Ref. [59] in the case of sparse ER graphs, the $\lambda = 0$ case is seen to be special. It also gives the strongly multifractal behaviour mentioned here, and is related to leaves of the network (i.e. low degree nodes). In Ref. [59] however, the "weakly multifractal" phase for $\lambda\neq 0$ is also explored. Is it also present here?

3- I think it is important to clarify some of the definitions used here. It would be helpful in particular to emphasize earlier and more clearly (i.e. perhaps in Section 2) that $\tau (q) = q-1$ for all q connotes extended states, that deviations from this behaviour indicate multifractality, and that $\tau(q) = 0$ for higher moments (i.e. q greater than some lower bound) indicates localization. Please correct me if I got any of the aforementioned items wrong. Incidentally, the behaviour in Eq. (44) is typical of Anderson localization on well-connected tree-like graphs, but I believe it fits the definition of "strongly multifractal" in Evers and Mirlin Mod Rev Phys (2008) in the same way that Eq. (56) does. Do the authors agree with this assessment?

4- The approximation given in Eq. (29) is tested against the average DoS of bona fide weighted configuration model Laplacians in Fig. 1. Thereafter, only Eq. (29) is used to check analytical results against numerics. However, it strikes me that a very similar approximation is made in the analytical procedure. Is this the case?

Regardless, perhaps it would be sensible also to check the results of direct diagonalization of configuration model networks for a more complicated observable such as the local DoS distribution at least once (it seems from Ref [59] that this should be within the realm of computational possibility).

5 - Related to this, I believe the "annealed network approximation", whereby one replaces the network by a weighted fully-connected graph with appropriately modified statistics (as is encapsulated by Eq. (29)), has already been given for adjacency matrices with $J_0 \neq 0$ and $J_1 \neq 0$ in Baron Phys Rev E (2022). The annealed network approximation has also been used in many other contexts (see e.g. Dorogovtsev, Goltsev, and Mendes Rev. Mod. Phys. (2008) or Carro, Toral, San Miguel Sci. Rep. (2016)). Although the use of the approximation to perform numerics in the case of Laplacian matrices is new here, I believe these related works should be referenced.

6- Is Eq. (46) what is called the Fyodorov distribution in Ref. [29], or is it different (aside from the shift due to the non-zero mean $J_0 \neq 0$)?

7- In general, it would be nice to see a bit more direct comparison to closely related results of previous works. That is, what are the main differences between the results (particularly $\tau(q)$ and $f(\alpha)$) seen here and those found in the case of (a) the adjacency matrix of sparse networks in Ref. [59] and (b) the dense but heterogeneous adjacency matrix in Ref [35]?

Recommendation

Ask for minor revision

  • validity: good
  • significance: high
  • originality: good
  • clarity: high
  • formatting: good
  • grammar: perfect

Author:  Fernando Metz  on 2024-11-25  [id 4989]

(in reply to Report 2 on 2024-10-14)

Referee: The manuscript by da Silva, Tapias, Sollich and Metz is a nice collaboration combining ideas
from the authors' previous works. Tapias and Sollich recently described the strongly multifractal states
around $\lambda=0$ for the adjacency matrix of weighted configuration model networks (notably following da
Silva and Metz in using the gamma distribution of rescaled degrees). Together with da Silva and Metz, the
ensemble of authors have now exploited some clever new tricks, in combination with the trusty cavity
method, to gain insight into the Laplacian matrix of such networks.

Referee: The work is clear, suitably pedagogical, and gives the reader a rather intuitive picture of why we
ought to expect novel behaviour of the eigenstates in this model: In the case with no off-diagonal
disorder, the Laplacian has only diagonal disorder (dictated by the degree distribution). Naturally, this
leads to localized states, with an average DoS dictated by the degree distribution. When all nodes have
the same degree but off-diagonal disorder is allowed, there is no diagonal disorder and the states are
extended, with a DoS that follows a shifted version of (what I think is) Fyodorov's result. When we have
both diagonal and off-diagonal disorder, one therefore expects that there could be an
interesting mix between the two extremes.

Referee: A thorough analysis is provided, showing the behaviour of the multifractal exponents, the singularity
spectrum and the local and average DoS in selected cases. Particular attention is paid to
the $\lambda=0$, where one finds a novel localization transition (as a function of the degree
heterogeneity) between localized multifractal states and non-ergodic delocalized states with weak
multifractality. This also coincides with a transition from singular to finite DoS at $\lambda=0$.

We thank the referee for the careful reading of our paper and for the positive comments regarding our research.

Referee: I would recommend publication after minor revision. I have only the following comments and queries.

Referee: Have the authors also considered the case of the normalized Laplacian (where the rows are
divided by the degree of the node in question, such that the diagonal elements are all equal to
unity -- see e.g. Ref [29] of the manuscript)? In many applications, it is the normalized version
that is used, but I can imagine that the phenomenology might differ quite a lot to that of the
version used here. Certainly, the DoS is different (one obtains the Wigner semi-circle in the
dense limit), and in the fully localized phase with $J_1=0$ the states would become
degenerate. I of course do not recommend that the same thorough analysis be carried out
for the normalized version, but a comment on this could be elucidating for readers.

We thank the referee for pointing this out. The referee is right that the normalized
Laplacian $\boldsymbol{\mathcal{L}}$ is important in certain applications, but
degree heterogeneities do not play any role in the spectral properties of
$\boldsymbol{\mathcal{L}}$ in the high-connectivity limit $c \rightarrow \infty$. To make this point
clear, let us obtain the form of $\boldsymbol{\mathcal{L}}$ for $c \rightarrow \infty$. The elements of the normalized Laplacian read
\begin{equation}
\mathcal{L}_{ij} = \delta_{ij} - (1 - \delta_{ij}) \frac{A_{ij}}{\sqrt{k_i k_j}},
\end{equation}
where $A_{ij} = c_{ij} J_{ij}$ are the elements of the weighted adjacency matrix. For simplicity, let us
consider that $J_{ij}$ has zero mean. Since the off-diagonal elements of $\mathcal{L}_{ij}$ are normalized
by the degrees $k_i$, we assume that the second moment of $J_{ij}$ is given
by $J_{1}^2$, independently of $c$ (this is different from our paper, see page 5). This scaling
ensures that $\boldsymbol{\mathcal{L}}$ converges to a finite limit as $c \rightarrow \infty$.
Thus, using the asymptotic form of $A_{ij}$ for $c \rightarrow \infty$, we find that $\mathcal{L}_{ij}$
converges to
\begin{equation}
\mathcal{L}_{ij} = \delta_{ij} - (1 - \delta_{ij}) \frac{J_1 g_{ij}}{\sqrt{N}},
\end{equation}
where $g_{ij}$ is a Gaussian random variable with mean zero and unit variance. Thus, the spectral
density of the normalized Laplacian converges to a shifted Wigner
law as $c \rightarrow \infty$, independently of the degree distribution, which makes the study of
the spectral properties of the normalized Laplacian on highly-connected networks
less interesting. We have included a brief comment about the normalized Laplacian after Eq. (29).

Referee: As is also discussed in Ref. [59] in the case of sparse ER graphs, the $\lambda=0$ case is
seen to be special. It also gives the strongly multifractal behaviour mentioned here, and is related
to leaves of the network (i.e. low degree nodes). In Ref. [59] however, the ``weakly multifractal'' phase
for $\lambda \neq 0$ is also explored. Is it also present here?

As the referee points out, we have also found a strong multifractal behavior in the case of
heterogeneous coupling strengths at $\lambda = 0$. For $\lambda \neq 0$, we have not
explored systematically the eigenvector statistics. However, we believe that, because of
the exponential tail of the LDoS distribution (see Figure 7 for $\lambda=2$), the eigenvectors
in this case are fully extended, and there is no ``weakly multifractal'' phase. We
have included this information in section 4.2.2.

Referee: I think it is important to clarify some of the definitions used here. It would be
helpful in particular to emphasize earlier and more clearly (i.e. perhaps in Section 2) that $\tau(q)=q-1$ for
all $q$ connotes extended states, that deviations from this behaviour indicate multifractality, and
that $\tau(q)=0$ for higher moments (i.e. $q$ greater than some lower bound) indicates localization. Please
correct me if I got any of the aforementioned items wrong. Incidentally, the behaviour in Eq. (44) is
typical of Anderson localization on well-connected tree-like graphs, but I believe it fits the definition
of "strongly multifractal" in Evers and Mirlin Mod Rev Phys (2008) in the same way that Eq. (56)
does. Do the authors agree with this assessment?

We thank the referee for these comments. Following the referee's suggestion, we have briefly
explained the behaviour of $\tau(q)$ in the limiting situations of fully delocalized
and localized states after Eq. (8). Additionally, we agree with the statement posed by the referee, namely, that
both equations (44) and (56) are examples of localized eigenvectors with strong multifractal
behavior, as discussed in [Evers and Mirlin Mod Rev Phys (2008)]. We have slightly modified the text
after Eq. (44) to emphasize the multifractal nature of the eigenvectors also for homogeneous coupling strengths.

Referee: The approximation given in Eq. (29) is tested against the average DoS of bona fide weighted
configuration model Laplacians in Fig. 1. Thereafter, only Eq. (29) is used to check analytical results
against numerics. However, it strikes me that a very similar approximation is made in the
analytical procedure. Is this the case?

It is not clear what the referee means by "a very similar approximation" in this context. In the
analytical procedure, explained in appendix A, we expand the characteristic functions associated with
the distributions $\mathcal{W}_z$ and $\mathcal{V}_z$ of the self-energies
$S(z) = \sum_{\ell=1}^k J_\ell \left[ 1-J_\ell g_\ell(z) \right]^{-1}$ in powers of $1/c$ (see Eqs. (A.2) and (A.6)).
Since the first two moments of $p_J$ are of $\mathcal{O}(1/c)$, this expansion produces quadratic forms
in the exponents of the characteristic functions, which allows us to analytically compute the
Fourier transforms that yield the expressions for $\mathcal{W}_z$ and $\mathcal{V}_z$. Therefore, we do
not explicitly use Eq. (29) in the analytical calculations of appendix A.

Referee: Regardless, perhaps it would be sensible also to check the results of direct diagonalization of
configuration model networks for a more complicated observable such as the local DoS distribution at
least once (it seems from Ref [59] that this should be within the realm of computational possibility).

We thank the referee for the suggestion. We have included in Fig. 3 a comparison between the LDoS distribution
obtained from the configuration model with the theoretical results given by Eq. (33) for $\lambda=0.5$.

Referee: Related to this, I believe the "annealed network approximation", whereby one replaces the
network by a weighted fully-connected graph with appropriately modified statistics (as is
encapsulated by Eq. (29)), has already been given for adjacency matrices with $J_0\neq 0$ and
$J_1\neq 0$ in Baron Phys Rev E (2022). The annealed network approximation has also been used in
many other contexts (see e.g. Dorogovtsev, Goltsev, and Mendes Rev. Mod. Phys. (2008) or Carro, Toral,
San Miguel Sci. Rep. (2016)). Although the use of the approximation to perform numerics in the case
of Laplacian matrices is new here, I believe these related works should be referenced.

We will cite these references below Eq. (29).

Referee: Is Eq. (46) what is called the Fyodorov distribution in Ref. [29], or is
it different (aside from the shift due to the non-zero mean $J_0\neq 0$)?

The referee is right, Eq. (46) corresponds to what is referred to as Fyodorov distribution
in reference [29]. The spectral density $\rho(\lambda)$ in [29] is proportional to the real part
of the following function
\begin{equation}
F(z) = iz + \sqrt{\frac{\pi}{2}}\mathrm{erfc}\bigg(\frac{F(z)}{\sqrt{2}}\bigg)e^{F^2(z)/2},
\end{equation}
where $z=\lambda - i\epsilon.$ By using the definition of the Faddeeva function
(see Eq. (27)), one can write Eq. (46) as
\begin{equation*}
\langle G\rangle = i \, \sqrt{\frac{\pi}{2}} \, \mathrm{erfc}\bigg[ \frac{-i(\langle G\rangle -z)}{\sqrt{2}} \bigg] e^{-(\langle G\rangle -z)^2/2},
\end{equation*}
where we have set $J_0=0$ and $J_1=1$ for simplicity. If we define $\langle G\rangle = iF(z) +z$ and
substitute this relation in the above equation, we recover the above self-consistent
equation for $F(z)$. In the limit $\epsilon \to 0^+$, the spectral density follows from
$\mathrm{Re}({F(z)}) = \mathrm{Im}{\langle G\rangle}$. We have mentioned that Eq. (46) is
equivalent to the spectral density in reference [16] after Eq. (46).

Referee: In general, it would be nice to see a bit more direct comparison to closely related
results of previous works. That is, what are the main differences between the results (particularly $\tau(q)$
and $f(\alpha)$ seen here and those found in the case of (a) the adjacency matrix of sparse networks in Ref. [59] and (b) the
dense but heterogeneous adjacency matrix in Ref [35])?

We have included an extra appendix with a summary of the results for the singularity spectrum and
the multifractal exponents in the case of the adjacency matrix (references [35] and [59]). We have
also modified the text after Eq. (58) to refer to appendix D and emphasize the comparison with reference [59].

---

## Round 2 · Author Response

Dear editors,

We resubmit hereby the manuscript entitled "Spectral properties, localization transition and multifractal eigenvectors of the Laplacian on heterogeneous networks" for publication in SciPost Physics. We have answered to all points raised by the referees and the paper has been modified accordingly. We hope that the resubmitted version of the manuscript will be accepted for publication.

Sincerely,
Fernando Metz
(corresponding author)

---

## Round 2 · List of Changes

All changes in the manuscript have been specified in the replies to the referees.

---

## Editorial Decision

accepted_in_target_journal